# *Squeeze*, *Recover* and *Relabel*: Dataset Condensation at ImageNet Scale From A New Perspective

**Zeyuan Yin**[*1], **Eric Xing**[1,2], **Zhiqiang Shen**[*1]
[1]Mohamed bin Zayed University of AI   [2]Carnegie Mellon University
{zeyuan.yin,eric.xing,zhiqiang.shen}@mbzuai.ac.ae

## Abstract

We present a new dataset condensation framework termed **S**queeze (), **Re**cover () and **Re**label () ($SRe^2L$) that decouples the bilevel optimization of model and synthetic data during training, to handle varying scales of datasets, model architectures and image resolutions for efficient dataset condensation. The proposed method demonstrates flexibility across diverse dataset scales and exhibits multiple advantages in terms of arbitrary resolutions of synthesized images, low training cost and memory consumption with high-resolution synthesis, and the ability to scale up to arbitrary evaluation network architectures. Extensive experiments are conducted on Tiny-ImageNet and full ImageNet-1K datasets[1]. Under 50 IPC, our approach achieves the highest 42.5% and 60.8% validation accuracy on Tiny-ImageNet and ImageNet-1K, outperforming all previous state-of-the-art methods by margins of 14.5% and 32.9%, respectively. Our approach also surpasses MTT [1] in terms of speed by approximately 52× (ConvNet-4) and 16× (ResNet-18) faster with less memory consumption of 11.6× and 6.4× during data synthesis. Our code and condensed datasets of 50, 200 IPC with 4K recovery budget are available at link.

## 1   Introduction

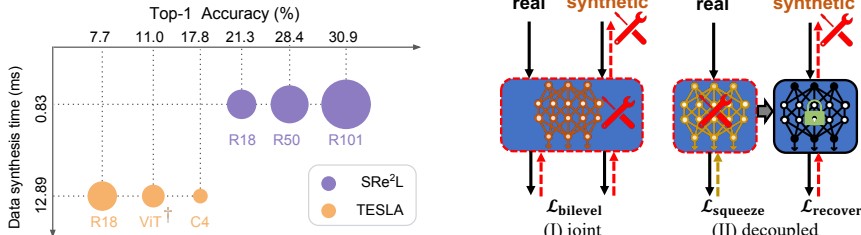

Figure 1: Left is data synthesis time vs. accuracy on ImageNet-1K with 10 IPC (Images Per Class). Models include ConvNet-4, ResNet-{18, 50, 101}. † indicates ViT with 10M parameters [2]. Right is the comparison of widely-used bilevel optimization and our proposed decoupled training scheme.

Over the past few years, the task of data condensation or distillation has garnered significant interest within the domain of computer vision [3, 4, 5, 6, 7, 1]. By distilling large datasets into representative, compact subsets, data condensation methods enable rapid training and streamlined storage, while preserving essential information from the original datasets. The significance of data condensation on both research and applications cannot be understated, as it plays a crucial role in the efficient handling

---

*Equal contribution. Project page: https://zeyuanyin.github.io/projects/SRe2L/.

[1]We discard small datasets like MNIST and CIFAR to highlight the scalability and capability of the proposed method on large-scale datasets for real-world applications.

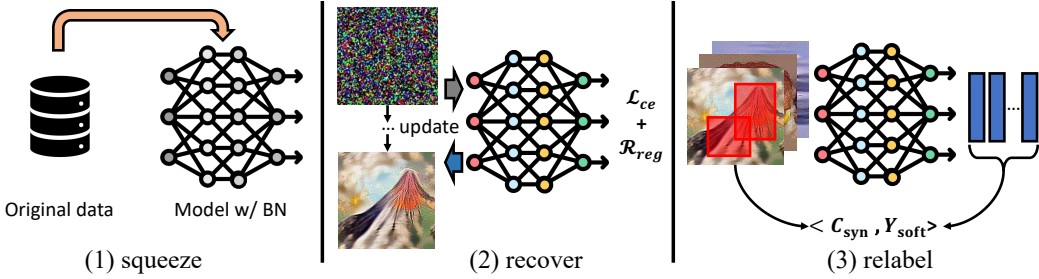

| (1) squeeze | (2) recover | (3) relabel |

Figure 2: Overview of our framework. It consists of three stages: in the first stage, a model is trained from scratch to accommodate most of the crucial information from the original dataset. In the second stage, a recovery process is performed to synthesize the target data from the Gaussian noise. In the third stage, we relabel the synthetic data in a crop-level scheme to reflect the true label of the data.

and processing of vast amounts of data across numerous fields. Through the implementation of sophisticated algorithms, such as Meta-Model Matching [3, 5], Gradient Matching [8, 9, 7], Distribution Matching [10, 6], and Trajectory Matching [1, 11], data condensation has made remarkable strides. However, prior solutions predominantly excel in distilling small datasets such as MNIST, CIFAR, Tiny-ImageNet [12], downscaled ImageNet [13] featuring low resolution, or a subset of ImageNet [7]. This limitation arises due to the prohibitive computational expense incurred from executing a massive number of unrolled iterations during the bilevel optimization process (comprising an inner loop for model updates and an outer loop for synthetic data updates). In our study, employing a meticulously designed decoupling strategy for model training and synthetic data updating (as illustrated in Fig. 1 left), the proposed method is capable of distilling the entire large-scale ImageNet dataset at the conventional resolution of $224 \times 224$ while maintaining state-of-the-art performance. Remarkably, our training/synthesis computation outstrips the efficiency of prior approaches, even those utilizing reduced resolution or subsets of ImageNet. A comparison of efficiency is provided in Table 1.

To address the huge computational and memory footprints associated with training, we propose a tripartite learning paradigm comprised of *Squeeze*, *Recover*, and *Relabel* stages. This paradigm allows for the decoupling of the condensed data synthesis stage from the real data input, as well as the segregation of inner-loop and outer-loop optimization. Consequently, it is not restricted by the scale of datasets, input resolution, or the size of network architectures. Specifically, in the initial phase, rather than simultaneous sampling of real and synthetic data and its subsequent network processing to update the target synthetic data, we bifurcate this process into *Squeeze* and *Recover* stages to distinguish the relationship and reduce similarity between real and synthetic data.

In the subsequent phase, we exclusively align the batch normalization (BN) [14] statistics derived from the model trained in the first phase to synthesize condensed data. In contrast to feature matching solutions that perform the matching process solely on individual batches within each iteration, the trained BN statistics on original data span the entire dataset, thereby providing a more comprehensive and accurate alignment between the original dataset and synthetic data. Given that real data is not utilized in this phase, the decoupled training can considerably diminish computational costs compared to the preceding bilevel training strategy. In the final phase, we employ the models trained in the first phase to relabel the synthetic data, serving as a data-label alignment process. An in-depth overview of this process is illustrated in Fig. 2.

**Advantages.** Our approach exhibits the following merits: **(1)** It can process large resolution condensation during training effortlessly with a justified computational cost. **(2)** Unlike other counterparts, we can tackle the relatively large datasets of Tiny-ImageNet and ImageNet-1K to make our method more practical for real-world applications. **(3)** Our approach can directly utilize many off-the-shelf pre-trained large models that contain BN layers, which enables to further save the training overhead.

Extensive experiments are performed on Tiny-ImageNet and ImageNet-1K datasets. On ImageNet-1K with $224 \times 224$ resolution and IPC 50, the proposed approach obtains a remarkable accuracy of 60.8%, outperforming all previous methods by a large margin. We anticipate that our research will contribute to the community's confidence in the practical feasibility of large-scale dataset condensation using a decoupled synthesis strategy from the real data, while maintaining reasonable computational costs.

| Methods | Condensed Arch | Time Cost (ms) | Peak GPU Memory Usage (GB) |
|---|---|---|---|
| DM [10] | ConvNet-4 | 18.11 | 10.7 |
| MTT [1] | ConvNet-4 | 12.64 | 48.9 |
| | ConvNet-4 | 0.24 | 4.2 |
| SRe$^2$L (Ours) | ResNet-18 | 0.75 | 7.6 |
| | ResNet-50 | 2.75 | 33.8 |

Table 1: **Synthesis Time and Memory Consumption** on Tiny-ImageNet ($64 \times 64$ resolution) using a single RTX-4090 GPU for all methods. Time Cost represents the consumption (ms) when generating one image with one iteration update on synthetic data. Peak value of GPU memory usage is measured or converted with a batch size of 200 (1 IPC as the dataset has 200 classes).

**Contributions.**

• We propose a new framework for large-scale dataset condensation, which involves a three-stage learning procedure of squeezing, recovery, and relabeling. This approach has demonstrated exceptional efficacy, efficiency and robustness in both the data synthesis and model training phases.

• We conduct a thorough ablation study and analysis, encompassing the impacts of diverse data augmentations for original data compression, various regularization terms for data recovery, and diverse teacher alternatives for relabeling on the condensed dataset. The comprehensive specifications of the learning process can offer valuable insights for subsequent investigations in the domain.

• To the best of our knowledge, this is the first work that enables to condense the full ImageNet-1K dataset with an inaugural implementation at a standard resolution of $224 \times 224$, utilizing widely accessible NVIDIA GPUs such as the 3090, 4090, or A100 series. Furthermore, our method attains the highest accuracy of 60.8% on full ImageNet-1K within an IPC constraint of 50 using justified training time and memory cost, outperforming all previous methods by a significant margin.

## 2   Approach

**Data Condensation/Distillation.** The objective of dataset condensation is to acquire a small synthetic dataset that retains a substantial amount of information present in the original data. Suppose we have a large labeled dataset $\mathcal{T} = \left\{ (\boldsymbol{x}_1, \boldsymbol{y}_1), \ldots, (\boldsymbol{x}_{|\mathcal{T}|}, \boldsymbol{y}_{|\mathcal{T}|}) \right\}$, we aim to learn a small condensed dataset $\mathcal{C}_{\text{syn}} = \left\{ (\widetilde{\boldsymbol{x}}_1, \widetilde{\boldsymbol{y}}_1), \ldots, (\widetilde{\boldsymbol{x}}_{|\mathcal{C}|}, \widetilde{\boldsymbol{y}}_{|\mathcal{C}|}) \right\}$ ($|\mathcal{C}| \ll |\mathcal{T}|$) that preserves the crucial information in original $\mathcal{T}$. The learning objective on condensed synthetic data is:

$$\boldsymbol{\theta}_{\mathcal{C}_{\text{syn}}} = \underset{\boldsymbol{\theta}}{\arg\min} \mathcal{L}_{\mathcal{C}}(\boldsymbol{\theta}) \tag{1}$$

where $\mathcal{L}_{\mathcal{C}}(\boldsymbol{\theta}) = \mathbb{E}_{(\widetilde{\boldsymbol{x}}, \widetilde{\boldsymbol{y}}) \in \mathcal{C}_{\text{syn}}} \left[ \ell(\phi_{\boldsymbol{\theta}_{\mathcal{C}_{\text{syn}}}}(\widetilde{\boldsymbol{x}}), \widetilde{\boldsymbol{y}}) \right]$, $\widetilde{\boldsymbol{y}}$ is the soft label coresponding to the synthetic data $\widetilde{\boldsymbol{x}}$.

The ultimate goal of data condensation task is to synthesize data for achieving a certain/minimal supremum performance gap on original evaluation set when models are trained on the synthetic data and the original full dataset, respectively. Following the definition of coresets [15] and $\epsilon$-approximate [16], the objective of data condensation task can be formulated as achieving:

$$\sup \left\{ \left| \ell\left(\phi_{\boldsymbol{\theta}_{\mathcal{T}}}(\boldsymbol{x}), \boldsymbol{y}\right) - \ell\left(\phi_{\boldsymbol{\theta}_{\mathcal{C}_{\text{syn}}}}(\boldsymbol{x}), \boldsymbol{y}\right) \right| \right\}_{(\boldsymbol{x}, \boldsymbol{y}) \sim \mathcal{T}} \leq \epsilon \tag{2}$$

where $\epsilon$ is the performance gap for models trained on the synthetic data and the original full dataset. Thus, we aim to optimize the synthetic data $\mathcal{C}_{\text{syn}}$ through:

$$\underset{\mathcal{C}_{\text{syn}}, |\mathcal{C}|}{\arg\min} \left( \sup \left\{ \left| \ell\left(\phi_{\boldsymbol{\theta}_{\mathcal{T}}}(\boldsymbol{x}), \boldsymbol{y}\right) - \ell\left(\phi_{\boldsymbol{\theta}_{\mathcal{C}_{\text{syn}}}}(\boldsymbol{x}), \boldsymbol{y}\right) \right| \right\}_{(\boldsymbol{x}, \boldsymbol{y}) \sim \mathcal{T}} \right) \tag{3}$$

Then, we can learn $<\text{data}, \text{label}> \in \mathcal{C}_{\text{syn}}$ with the associated number of condensed data in each class.

**Decoupling the condensed data optimization and the neural network training:** Conventional solutions, such as FRePo [5], CAFE [6], DC [8], typically choose for the simultaneous optimization of the backbone network and synthetic data within a singular training framework, albeit in an iterative fashion. The primary drawback associated with these joint methods is their computational burden due to the unrolling of the inner-loop during each outer-loop update, coupled with bias transference

from real to synthetic data as a result of truncated unrolling. The objective of this study is to devise an efficient learning framework capable of individually decoupling model training and synthetic data optimization. This approach circumvents information bias stemming from real data, concurrently enhancing efficiency in handling diverse scales of datasets, model architectures, and image resolutions, thereby bolstering effective dataset condensation. Our framework is predicated on the assumption that pivotal information within a dataset can be adequately trained and preserved within a deep neural network. The training procedure of our approach is elaborated in the following section.

## 2.1 Decoupling Outer-loop and Inner-loop Training

Inspired by recent advances in DeepDream [17], Inverting Image [18, 19] and data-free knowledge transfer [20], we propose a decoupling approach to disassociate the traditional bilevel optimization inherent to dataset condensation. This is accomplished via a tripartite process to reformulate it into a unilevel learning procedure.

**Stage-1 Squeeze ( ):** During this stage, our objective is to extract pertinent information from the original dataset and encapsulate it within the deep neural networks, evaluating the impact of various data augmentation techniques, training strategies, etc. Deep neural networks typically comprise multiple parametric functions, which transform high-dimensional original data (e.g., pixel space of images) into their corresponding low-dimensional latent spaces. We can exploit this attribute to abstract the original data to the pretrained model and then reconstruct them in a more focused manner, akin to DeepDream [17] and Inverting Images [18, 19]. It's noteworthy that the purpose of this stage is to extract and encapsulate critical information from the original dataset. Hence, excessive data augmentation resulting in enhanced performance does not necessarily lead to the desired models. This approach diverges from previous solutions that sample two data batches from the original large-scale dataset $\mathcal{T}$ and the learnable small synthetic dataset $\mathcal{C}$. The learning procedure can be simply cast as a regular model training process on the original dataset with a suitable training recipe:

$$\boldsymbol{\theta}_{\mathcal{T}} = \arg\min_{\boldsymbol{\theta}} \mathcal{L}_{\mathcal{T}}(\boldsymbol{\theta}) \tag{4}$$

where $\mathcal{L}_{\mathcal{T}}(\boldsymbol{\theta})$ typically uses cross-entropy loss as $\mathcal{L}_{\mathcal{T}}(\boldsymbol{\theta}) = \mathbb{E}_{(\boldsymbol{x}, \boldsymbol{y}) \in \mathcal{T}}[\boldsymbol{y} \log (\boldsymbol{p}(\boldsymbol{x}))]$.

**Enabling BN Layers in ViT for Recovering Process**: In contrast to distribution matching [10] that aligns the feature distributions of the original and synthetic training data in sampled embedding spaces, thus allowing for the use of a randomly initialized network, our matching mechanism is solely performed on the Batch Normalization (BN) layers using their statistical properties, akin to data-free knowledge transfer [20]. Unlike the feature matching solution which executes the matching process on individual batches within each iteration, the referential BN statistics are calculated over the entirety of the dataset, providing a more comprehensive and representative alignment with the original dataset. Our experiments empirically substantiate that BN-matching can significantly outperform the feature-matching method. BN layer is commonly used in ConvNet but is absent in ViT. To utilize both ConvNet and ViT for our proposed data condensation approach, we engineer a BN-ViT which replaces all LayerNorm by BN layers and adds additional BN layers in-between two linear layers of feed-forward network, as also utilized in [21]. This marks the first instance of broadening the applicability of the data condensation architecture from ConvNets to encompass ViTs as well.

**Stage-2 Recover ( ):** This phase involves reconstructing the retained information back into the image space utilizing class labels, regularization terms, and BN trajectory alignment. Unlike conforming to batch feature distributions or comprehensive parameter distributions, we solely track the distribution of BN statistics derived from the original dataset. The pairing of BN and predictive probability distribution restricts the optimization process to a singular level, thereby significantly enhancing scalability. By aligning the final classification and intermediary BN statistics (mean and variance), the synthesized images are compelled to encapsulate a portion of the original image distribution. The learning objective for this phase can be formulated as follows:

$$\arg\min_{\mathcal{C}_{\text{syn}}, |\mathcal{C}|} \ell\left(\phi_{\boldsymbol{\theta}_{\mathcal{T}}}(\widetilde{\boldsymbol{x}}_{\text{syn}}), \boldsymbol{y}\right) + \mathcal{R}_{\text{reg}} \tag{5}$$

where $\mathcal{R}_{\text{reg}}$ is the regularization term. $\phi_{\boldsymbol{\theta}_{\mathcal{T}}}$ is the model pre-trained in the first stage and it will be frozen in this stage, we solely optimize the $\widetilde{\boldsymbol{x}}_{\text{syn}}$ as a single-level training process. Following [18, 20], we discuss three regularizers that can be used and the ablation for them is provided in our experiments. The first two regularizers are image prior regularisers of $\ell_2$ and total variation (TV) proposed in [18]:

$$\mathcal{R}_{\text{prior}}(\widetilde{\boldsymbol{x}}_{\text{syn}}) = \alpha_{\text{tv}} \mathcal{R}_{\text{TV}}(\widetilde{\boldsymbol{x}}_{\text{syn}}) + \alpha_{\ell_2} \mathcal{R}_{\ell_2}(\widetilde{\boldsymbol{x}}_{\text{syn}}) \tag{6}$$

where $\mathcal{R}_{\ell_2} = \|\widetilde{\boldsymbol{x}}_{\text{syn}}\|_2$, this regularizer encourages image to stay within a target interval instead of diverging. $\mathcal{R}_{\text{TV}} = \sum_{i,j} \left( (\widetilde{\boldsymbol{x}}_{i,j+1} - \widetilde{\boldsymbol{x}}_{ij})^2 + (\widetilde{\boldsymbol{x}}_{i+1,j} - \widetilde{\boldsymbol{x}}_{ij})^2 \right)^{\frac{\beta}{2}}$ where $\beta$ is used to balance the sharpness of the image with the removal of "spikes" and smooth the synthetic data. While, this is not necessary for dataset condensation since we care more about information recovery.

**Learning Condensed Data with BN Consistency**: DeepInversion [20] utilizes the feature distribution regularization term to improve the quality of the generated images. Here, we also leverage this property as our recovering loss term. It can be formulated as:

$$\begin{aligned} \mathcal{R}_{\text{BN}}(\widetilde{\boldsymbol{x}}) &= \sum_l \left\| \mu_l(\widetilde{\boldsymbol{x}}) - \mathbb{E}\left( \mu_l \mid \mathcal{T} \right) \right\|_2 + \sum_l \left\| \sigma_l^2(\widetilde{\boldsymbol{x}}) - \mathbb{E}\left( \sigma_l^2 \mid \mathcal{T} \right) \right\|_2 \\ &\approx \sum_l \left\| \mu_l(\widetilde{\boldsymbol{x}}) - \mathbf{BN}_l^{\text{RM}} \right\|_2 + \sum_l \left\| \sigma_l^2(\widetilde{\boldsymbol{x}}) - \mathbf{BN}_l^{\text{RV}} \right\|_2 \end{aligned} \tag{7}$$

where $l$ is the index of BN layer, $\mu_l(\widetilde{\boldsymbol{x}})$ and $\sigma_l^2(\widetilde{\boldsymbol{x}})$ are mean and variance. $\mathbf{BN}_l^{\text{RM}}$ and $\mathbf{BN}_l^{\text{RV}}$ are running mean and running variance in the pre-trained model at $l$-th layer, which are globally counted.

**Multi-crop Optimization**: `RandomResizedCrop`, a frequently utilized technique in neural network training, serves as a preventative measure against overfitting. Inspired by this, we propose the strategy of multi-crop optimization during the process of image synthesis with the aim of enhancing the informational content of the synthesized images. In practice, we implement it by randomly cropping on the entire image and subsequently resizing the cropped region to the target dimension of $224 \times 224$. Under these circumstances, only the cropped region is updated during each iteration. This approach aids in refining the recovered data when viewed from the perspective of cropped regions.

**Stage-3 Relabel (✂)**: To match our multi-crop optimization strategy, also to reflect the true soft label for the recovered data. We leverage the pre-generated soft label approach as FKD [22].

$$\widetilde{\boldsymbol{y}}_i = \phi_{\boldsymbol{\theta}_{\mathcal{T}}}(\widetilde{\boldsymbol{x}}_{\mathbf{R}_i}) \tag{8}$$

where $\widetilde{\boldsymbol{x}}_{\mathbf{R}_i}$ is the $i$-th crop in the synthetic image and $\widetilde{\boldsymbol{y}}_i$ is the corresponding soft label. Finally, we can train the model $\phi_{\boldsymbol{\theta}_{\mathcal{C}_{\text{syn}}}}$ on the synthetic data using the following objective:

$$\mathcal{L}_{\text{syn}} = -\sum_i \widetilde{\boldsymbol{y}}_i \log \phi_{\boldsymbol{\theta}_{\mathcal{C}_{\text{syn}}}}(\widetilde{\boldsymbol{x}}_{\mathbf{R}_i}) \tag{9}$$

We found that this stage is crucial to make synthetic data and labels more aligned and also significantly improve the performance of the trained models.

**Discussion: How does the proposed approach reduce compute and memory consumption?** Existing solutions predominantly employ bilevel optimization [3, 23, 4] or long-range parameter matching strategies [1, 11], which necessitate the feeding of real data into the network to generate guiding variables (e.g., features, gradients, etc.) for target data updates, as well as for the backbone network training, through an iterative process. These approaches incur considerable computational and memory overhead due to the concurrent presence of real and synthetic data on computational hardware such as GPUs, thereby rendering this training strategy challenging to scale up for larger datasets and models. To this end, a natural idea is to decouple real and synthetic data during the training phase, thereby necessitating only minimal memory during each training session. This is achieved by bifurcating the bilevel training into a two-stage process: squeezing and recovering. Moreover, we can conveniently utilize off-the-shelf pre-trained models for the first squeezing stage.

## 3 Experiments

In this section, we evaluate the performance of our proposed SRe$^2$L over various datasets, models and tasks. First, we conduct extensive ablation experiments to investigate the effect of each component in three stages. Next, we demonstrate the superior results of SRe$^2$L in large-scale datasets, cross-architecture generalization, and continual learning application. Finally, we provide the comparison of visualizations on distilled data with other state-of-the-art methods.

**Experiment Setting.** We evaluate our method SRe$^2$L on two large-scale datasets Tiny-ImageNet [12] and full ImageNet-1K [24]. Detailed comparison among variants of ImageNet-1K is in appendix. For backbone networks, we employ ResNet-{18, 50, 101} [25], ViT-Tiny [26], and our new constructed BN-ViT-Tiny (Sec. 2.1) as the target model training. For distilling ImageNet-1K, we

| Recovery model | Squeezing Budget (iteration) | | | | Data Augmentation | |
|---|---|---|---|---|---|---|
| | 50 | 100 | 200 | 400 | Mixup | CutMix |
| ResNet-18 | 37.88 | 37.49 | 32.18 | 23.88 | 35.59 | 25.90 |
| ResNet-50 | 39.19 | 33.63 | 30.65 | 26.43 | 36.95 | 29.61 |

Table 2: Ablation of squeezing budget and data augmentation on Tiny-ImageNet.

recover the data from PyTorch off-the-shelf pre-trained ResNet-{18, 50} with the Top-1 accuracy of {69.76%, 76.13%} for saving re-training computational overhead. For distilling Tiny-ImageNet, ResNet-{18, 50} are used as base models, while the first 7×7 Conv layer is replaced by 3×3 Conv layer and the maxpool layer is discarded, following MoCo (CIFAR) [27]. After that, they are trained from scratch on Tiny-ImageNet. More implementation details are provided in appendix.

**Evaluation and baselines.** Following previous works [3, 1, 8], we evaluate the quality of condensed datasets by training models from scratch on them and report the test accuracy on real val datasets.

## 3.1 Squeezing Analysis

Numerous training methods exist to enhance the accuracy of models [28], including extended training cycles/budgets and data augmentation strategies such as Mixup [29] and CutMix [30]. We further examine the performance of synthetic data regenerated from models demonstrating varied levels of accuracy. This investigation is aimed at addressing a compelling question: *Does a model with superior squeezing ability yield more robust recovered data?* Here, "a model with superior squeezing ability" is defined as a model that exhibits strengthened accuracy on the validation set.

**Squeezing Budget.** In Table 2, we observe a decrement in the performance of the recovered model as the squeezing budget escalates. This suggests an increasing challenge in data recovery from a model that has been trained over more iterations. Consequently, we adopt a squeezing budget of 50 iterations as the default configuration for our experiments on Tiny-ImageNet.

**Data Augmentation.** Table 2 also shows that data augmentation methods in the squeeze stage decrease the final accuracy of the recovered data. In summary, results on Tiny-ImageNet indicate that extending the training duration and employing data augmentation during the squeeze phase exacerbates the complexity of data recovery from the squeezed model.

## 3.2 Recovering Analysis

The condensed data are crafted and subsequently relabeled utilizing a pre-trained ResNet-18 with a temperature of 20. Then, we report the val performance of training a ResNet-18 from scratch.

**Image Prior Regularization.** $\mathcal{R}_{TV}$ and $\mathcal{R}_{\ell_2}$ are extensively applied in image synthesis methods [18, 31]. However, in the pursuit of extracting knowledge from the pre-trained model, our focus is predominantly on the recuperation of semantic information as opposed to visual information. Analyzing from the standpoint of evaluation performance as shown in appendix, $\mathcal{R}_{\ell_2}$ and $\mathcal{R}_{TV}$ barely contribute to the recovery of image semantic information, and may even serve as impediments to data recovery. Consequently, these image prior regularizations are omitted during our recovery phase.

**Multi-crop Optimization.** To offset the `RandomResizedCrop` operation applied to the training data during the subsequent model training phase, we incorporate a corresponding `RandomResizedCrop` augmentation on synthetic data during recovery. This implies that only a minor cropped region in the synthetic data undergoes an update in each iteration. Our experimentation reveals that the multi-crop optimization strategy facilitates a notable improvement in validation accuracy, as in Appendix Table 8.

A comparative visualization and comparison with other non-crop settings is shown in Fig. 3. In the last column (SRe²L), multiple miniature regions enriched with categorical features spread across the entire image. Examples include multiple volcanic heads, shark bodies, bee fuzz, and mountain ridges. These multiple small feature regions populate the entire image, enhancing its expressiveness in terms of visualization. Therefore, the cropped regions on our synthetic images are not only more closely associated with the target categories but also more beneficial for soft label model training.

| Dataset | Recovery model | Recovering Budget | | | | Time (ms) |
|---|---|---|---|---|---|---|
| | | $0.5k$ | $1k$ | $2k$ | $4k$ | |
| Tiny-ImageNet | ResNet-18 | 34.82 | 37.88 | 41.35 | 41.11 | 0.75 |
| | ResNet-50 | 35.92 | 39.19 | 39.59 | 39.29 | 2.75 |
| ImageNet-1K | ResNet-18 | 38.94 | 43.69 | 46.71 | 46.73 | 0.83 |
| | ResNet-50 | 21.38 | 28.69 | 33.20 | 35.41 | 2.62 |

Table 3: Top-1 validation accuracy of ablation using ResNet-{18, 50} as recovery model with different updating iterations, and ResNet-18 as student model. "Time" represents the consuming time (ms) when training 1 image per iteration on one single NVIDIA 4090 GPU.

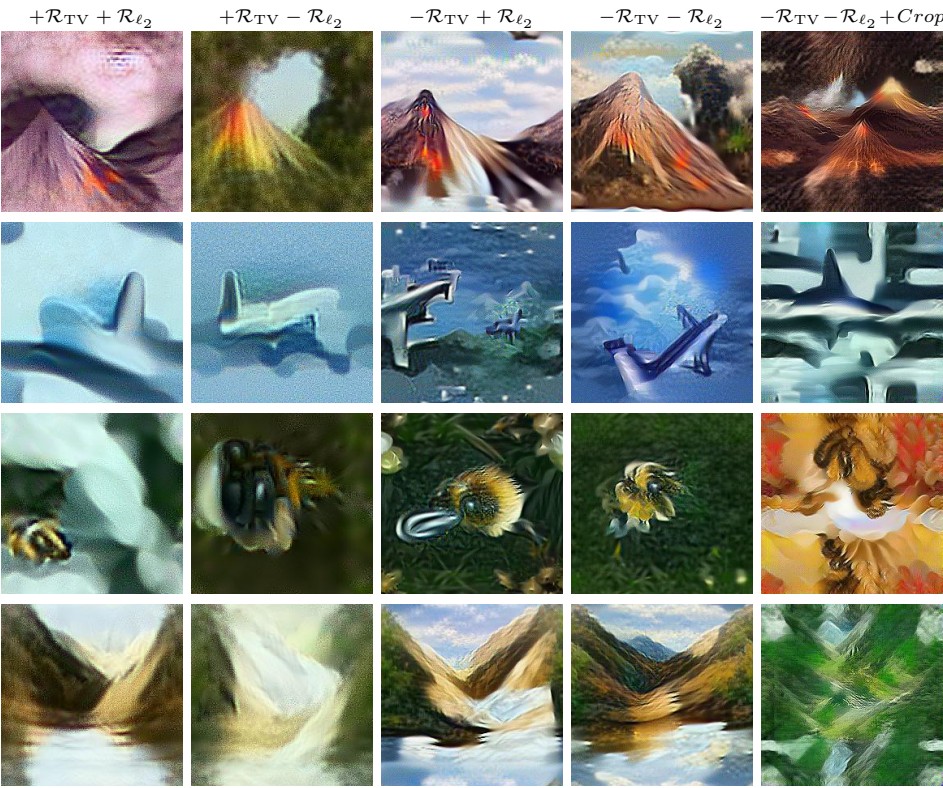

Figure 3: Visualization of distilled examples on ImageNet-1K under various regularization terms and crop augmentation settings. Selected classes are {Volcano, Hammerhead Shark, Bee, Valley}.

**Recover Budget.** We conduct various ablation studies to evaluate the impact of varying recovery budgets on the quality of synthetic data. The recovery budgets are designated as $[0.5k, 1k, 2k, 4k]$. As in Table 3, it indicates that employing a longer recovery budget on the same model results in superior classification accuracy on the recovered data. In the case of recovery from diverse models, the results demonstrate that the data recovery under the same iterative setting and budget from ResNet-50 is inferior to that from ResNet-18 on both datasets. This suggests that the process of data recovery from larger pre-trained models is more challenging on large-scale datasets, necessitating more iterations to ensure that the recovered data achieves comparable performance on downstream classification tasks. To strike a balance between performance and time cost, we impose a recovery budget of $1k$ iterations on Tiny-ImageNet and $2k$ iterations on ImageNet-1K for the ablations, and $4k$ for the best in Table 4.

## 3.3 Relabeling Analysis

We further study the influence of varying relabeling models and distinct softmax temperatures on different architectural models being optimized. In Fig. 4, we present three subfigures that represent

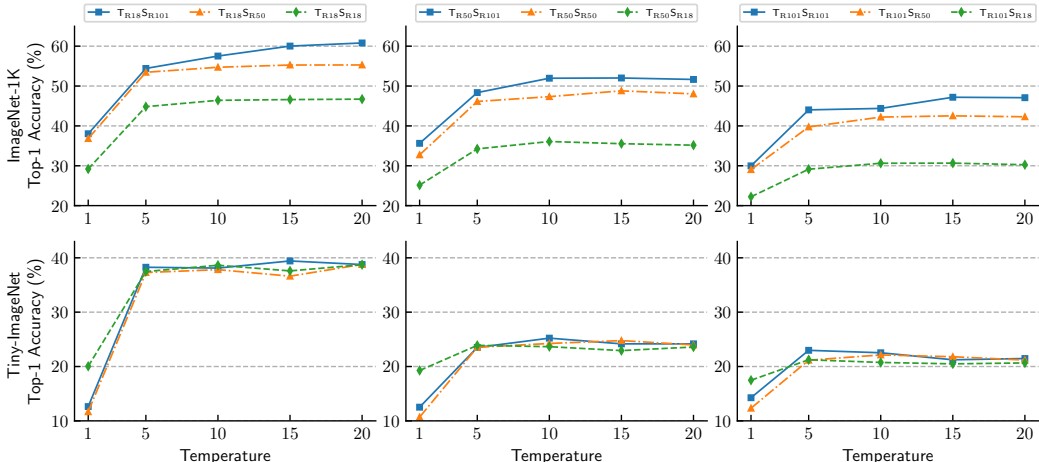

Figure 4: Top-1 val accuracy of models trained on various labels and temperature settings under IPC 50. T and S represent the reference model for relabeling and the target model to be trained, separately. R18, R50, and R101 are the abbreviation of ResNet-18, ResNet-50, and ResNet-101.

the training accuracy of the models on labels generated by three pre-trained models: ResNet-{18, 50, 101}. The training data utilized for these experiments is the same, which is recovered from a pre-trained ResNet-18 model.

**Relabeling Model.** In each row of Fig. 4 (i.e., on the same dataset), the results of three subfigures indicate that a smaller-scale teacher which is close or identical to the recovery model architecture always achieves better accuracy across ResNet-18, ResNet-50 and ResNet-101. Thus, it can be inferred that the labels of the recovered data are most accurate when the relabeling model aligns with the recovery model. Moreover, Top-1 errors tend to escalate with increasing disparity between the relabeling model and the recovery model. Consequently, in our three-stage methodology, we opt to employ an identical model for both recovery and relabeling processes.

**Temperature on Soft Label.** We conduct experiments encompassing five different temperature selections [1, 5, 10, 15, 20] specifically for label softmax operations under distillation configuration as [22]. The result indicates that the Top-1 accuracy experiences a rapid surge initially and subsequently plateaus when the temperature setting exceeds 10. The maximum Top-1 accuracy is recorded as $60.81\%$ when the temperature is fixed at 20, employing ResNet-18 as the teacher model and ResNet-101 as the student model. This observation underscores the beneficial effect of a higher temperature setting for label softmax operations in the training of the student model. Consequently, we elect to utilize a temperature value of 20 in our subsequent evaluation experiments.

**Model Training.** Contrary to previous data condensation efforts, where different architectures could not be effectively trained on the condensed data due to the mismatch of recovery models or overfitting on the limited synthetic data scale, our condensed data demonstrate the informativeness and training scalability as the inherent properties in real data. In each subfigure within Fig. 4, we observe a progressive increase in accuracy when training ResNet-18, ResNet-50, and ResNet-101 as the student networks. This indicates that our condensed data does not suffer from overfitting recovered from the squeezing model, and during inference, the Top-1 accuracy is consistently increasing using a network with enhanced capabilities.

### 3.4 Condensed Dataset Evaluation

**Tiny-ImageNet.** The results derived from the Tiny-ImageNet dataset are presented in the first group of Table 4. Upon evaluation, it can be observed that MTT achieves 28.0% under the IPC 50. In contrast, our results on ResNet-{18, 50, 101} architectures are 41.1%, 42.2%, and 42.5%, respectively, which significantly surpass the performance of MTT. A noteworthy observation is that our stronger backbones not only accomplish superior accuracy but are also robust for different recovery architectures. In light of the results under IPC 50 and 100 settings on the relatively small Tiny-ImageNet dataset, it is apparent that the larger backbone did not yield a proportional enhancement in performance from ResNet-18 to ResNet-101 by our method in Table 4 and Fig. 4. While, this is different from the observation on full ImageNet-1K that we discuss elaborately below.

| Dataset | IPC | MTT [1] | TESLA [11] | TESLA [11] (ViT) | TESLA [11] (R18) | SRe²L (R18) | SRe²L (R50) | SRe²L (R101) |
|---|---|---|---|---|---|---|---|---|
| Tiny-IN | 50 | 28.0±0.3 | - | - | - | 41.1±0.4 | 42.2±0.5 | 42.5±0.2 |
| | 100 | - | - | - | - | 49.7±0.3 | 51.2±0.4 | 51.5±0.3 |
| IN-1K | 10 | 64.0±1.3† | 17.8±1.3 | 11.0±0.2 | 7.7±0.1 | 21.3±0.6 | 28.4±0.1 | 30.9±0.1 |
| | 50 | - | 27.9±1.2 | - | - | 46.8±0.2 | 55.6±0.3 | 60.8±0.5 |
| | 100 | - | - | - | - | 52.8±0.3 | 61.0±0.4 | 62.8±0.2 |
| | 200 | - | - | - | - | 57.0±0.4 | 64.6±0.3 | 65.9±0.3 |

Table 4: Comparison with baseline models. † indicates the ImageNette dataset, which contains only 10 classes. TESLA [11] uses the downsampled ImageNet-1K dataset. Our results are derived from the full ImageNet-1K, which is more challenging on computation and memory, meanwhile, presenting greater applicability potential in real-world scenarios. The recovery model used in the table is R18.

| Squeezed Model | Evaluation Model | | | |
|---|---|---|---|---|
| | DeiT-Tiny | ResNet-18 | ResNet-50 | ResNet-101 |
| DeiT-Tiny-BN | 25.36 | 24.69 | 31.15 | 33.16 |
| ResNet-18 | 15.41 | 46.71 | 55.29 | 60.81 |

Table 5: ImageNet-1K Top-1 on Cross-Archtecture Generalization. Two recovery/squeezed models are used: DeiT-Tiny-BN and ResNet-18. Four evaluation models: DeiT-Tiny, ResNet-{18, 50, 101}.

**ImageNet-1K.** As shown in the second group of Table 4, employing the same model architecture of ResNet-18 under IPC 10, our approach improves the performance of TESLA from a baseline of 7.7% to a significance of 21.3%. Contrary to TESLA, where the performance deteriorates with larger model architectures, our proposed approach capitalizes on larger architectures, displaying an appreciable proportional performance enhancement. This indicates significant promise in the contemporary era of large-scale models. On IPC 50, 100 and 200, our method obtains consistent boosts on accuracy.

### 3.5 Cross-Architecture Generalization

It is important to verify the generalization property of our condensed datasets, ensuring its ability to effectively generalize to new architectures that it has not encountered during the synthesis phase. Fig. 4 and Table 5 demonstrate that our condensed dataset exhibits proficient cross-model generalization across ResNet-{18, 50, 101} and ViT-T. The results reveal that our condensed datasets maintain robustness across disparate and larger architectures. However, we observed suboptimal performance of ViT on the condensed dataset, potentially due to the model's inherent need for substantial training data as introduced in [26].

### 3.6 Synthetic Image Visualization

Fig. 5 provides a visual comparison of selected synthetic images from our condensed dataset and the corresponding images from the MTT condensed dataset. The synthetic images generated by our method manifest a higher degree of clarity on semantics, effectively encapsulating the attributes and contours of the target class. In

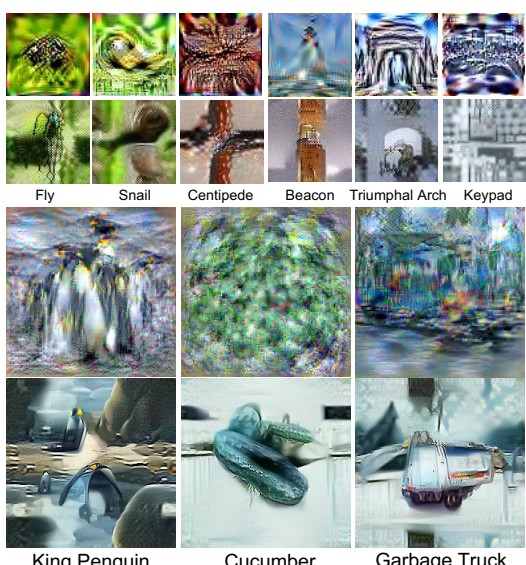

Figure 5: Visualization of MTT [1] and our SRe²L. The upper two rows are synthetic Tiny-ImageNet and the lower two rows are synthetic ImageNet-1K (the first row is MTT and second is ours).

contrast, synthetic images from MTT appear considerably blurred, predominantly capturing color information while only encapsulating minimal details about the target class. Consequently, SRe²L produces superior-quality images that not only embed copious semantic information to augment validation accuracy but also demonstrate superior visual performance.

### 3.7 Application: Continual Learning

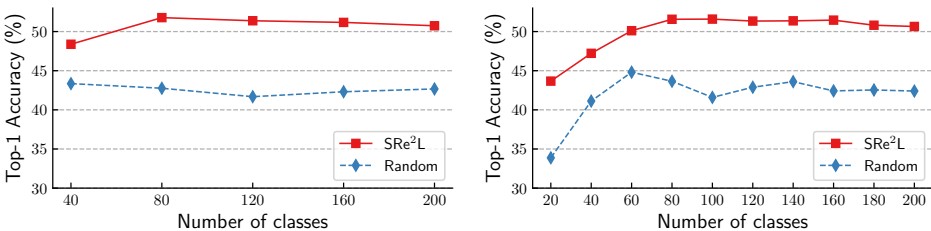

Figure 6: 5-step and 10-step class-incremental learning on Tiny-ImageNet.

Many prior studies [10, 5, 32, 7] have employed condensed datasets for continual learning to assess the quality of the synthetic data. We adhere to the method outlined in DM [10] class-incremental learning implementation, which is based on GDumb [33]. This method sequentially stores prior training data in memory and utilizes both new training data and stored data to learn a model from scratch. To demonstrate the superiority of our method in handling large-scale data, we conduct class incremental learning on Tiny-ImageNet, incorporating an escalating memory budget of 100 images per class, and training with ResNet-18. Fig. 6 shows both 5-step and 10-step class-incremental learning strategies, which partition 200 classes into either 5 or 10 learning steps, accommodating 40 and 20 classes per step respectively. Our results are clearly better than the baselines.

## 4 Related Work

Data condensation or distillation aims to create a compact synthetic dataset that preserves the essential information in the large-scale original dataset, making it easier to work with and reducing training time while achieving the comparable performance to the original dataset. Previous solutions are mainly divided into four categories: *Meta-Model Matching* optimizes for the transferability of models trained on the condensed data and uses an outer-loop to update the synthetic data when generalized to the original dataset with an inner-loop to train the network, methods include DD [3], KIP [23], RFAD [4], FRePo [5] and LinBa [34]; *Gradient Matching* performs a one-step distance matching process on the network trained on original dataset and the same network trained on the synthetic data, methods include DC [8], DSA [32], DCC [9] and IDC [7]; *Distribution Matching* directly matches the distribution of original data and synthetic data with a single-level optimization, methods include DM [10], CAFE [6], HaBa [35], IT-GAN [36], KFS [9]; *Trajectory Matching* matches the training trajectories of models trained on original and synthetic data in multiple steps, methods include MTT [1] and TESLA [11]. Our proposed decoupling method presents a new perspective for tackling this task, while our BN-matching recovering procedure can also be considered as a special format of *Distribution Matching* scheme on the synthetic data and global BN statistics distributions.

## 5 Conclusion

We have presented a novel three-step process approach for the dataset condensation task, providing a more efficient and effective way to harness the power of large-scale datasets. By employing the sequential steps of squeezing, recovering, and relabeling, this work condenses the large-scale ImageNet-1K while retaining its essential information and performance capabilities. The proposed method outperforms existing state-of-the-art condensation approaches by a significant margin, and has a wide range of applications, from accelerating the generating and training process to enabling the method that can be used in resource-constrained environments. Moreover, the study demonstrates the importance of rethinking conventional methods of data condensation and model training, as new solutions can lead to improvements in both computational efficiency and model performance. As the field of data condensation continues to evolve, the exploration of targeting approaches, such as the one presented in this work, will be crucial for the development of future condensation approaches that are more efficient, robust, and capable of handling vast amounts of data in a sustainable manner.

**Limitation and Future Work**: At present, a performance disparity persists between the condensed dataset and the original full dataset, indicating that complete substitution of the full data with condensed data is yet to be feasible. Another limitation is the extra storage for soft labels. Moving forward, our research endeavors will concentrate on larger datasets such as the condensation of ImageNet-21K, as well as other data modalities encompassing language and speech.

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

# Appendix

In the appendix, we provide details omitted in the main text, including:

## A   Implementation Details

### A.1   Dataset Statistics

Table 6 enumerates various permutations of ImageNet-1K training set, delineated according to their individual configurations. Tiny-ImageNet [12] incorporates 200 classes derived from ImageNet-1K, with each class comprising 500 images possessing a resolution of 64×64. ImageNette/ImageWoof [37] (alternatively referred to as subsets of ImageNet) include 10 classes from analogous subcategories, with each image having a resolution of 112×112. The MTT [1] framework introduces additional 10-class subsets of ImageNet, encompassing ImageFruit, ImageSquawk, Image-Meow, ImageBlub, and ImageYellow. ImageNet-10/100 [38] samples 10/100 classes from ImageNet while maintaining an image resolution of 224×224. Downsampled ImageNet-1K rescales the entirety of ImageNet data to a resolution of 64×64. In our experiments, we choose two standard datasets of relatively large scale: Tiny-ImageNet and the full ImageNet-1K.

| Training Dataset | #Class | #Img per class | Resolution | Method |
|---|---|---|---|---|
| Tiny-ImageNet [12] | 200 | 500 | 64×64 | MTT [1], FRePo [5], DM [10], SRe$^2$L (Ours) |
| ImageNette/ImageWoof [37] | 10 | ∼1,000 | 112×112 | MTT [1], FRePo [5] |
| ImageNet-10/100 [38] | 10/100 | ∼1,200 | 224×224 | IDC [7] |
| Downsampled ImageNet-1K [13] | 1,000 | ∼1,200 | 64×64 | TESLA [11], DM [10] |
| Full ImageNet-1K [24] | 1,000 | ∼1,200 | 224×224 | SRe$^2$L (Ours) |

Table 6: Variants of ImageNet-1K training set with different configurations.

### A.2   Squeezing Details

**Data Augmentation.** Table 2 in the main paper illustrates that the utilization of data augmentation approaches during the squeezing phase contributes to a decrease in the final accuracy of the recovered data. To summarize, the results on Tiny-ImageNet indicate that lengthening the training budget and the application of more data augmentations in the squeezing phase intensify the intricacy involved in data recovery from the compressed model, which is not desired.

Parallel conclusions are inferred from the compressed models for the ImageNet-1K dataset. For our experimental setup, we aimed to extract data from a pre-trained ResNet-50 model with available *V1* and *V2* weights in the PyTorch model zoo. The results propose that the task of data extraction poses a greater challenge from the ResNet-50 model equipped with *V2* weights as compared to the model incorporating *V1* weights. This can be attributed to the fact that models utilizing *V1* weights are trained employing a rudimentary recipe, whereas models with *V2* weights encompass numerous training enhancements, such as more training budget and data augmentations, to achieve cutting-edge performance. It is observed that these additional complexities impede the data recovery process even the pre-trained models are stronger. Therefore, the pre-trained models we employ for the recovery of ImageNet-1K are those integrating *V1* weights from the PyTorch model zoo.

**Hyper-parameter Setting.** We provide hyper-parameter settings for the two datasets in detail.

• Tiny-ImageNet: We train modified ResNet-{18, 50} models on Tiny-ImageNet data with the parameter setting in Table 7a. The well-trained ResNet-{18, 50} models achieve Top-1 accuracy of {59.47%, 61.17%} under the 50 epoch training budget.

| config | value |
|---|---|
| optimizer | SGD |
| base learning rate | 0.2 |
| weight decay | 1e-4 |
| optimizer momentum | 0.9 |
| batch size | 256 |
| learning rate schedule | cosine decay |
| training epoch | 100 |
| augmentation | RandomResizedCrop |

(a) Tiny-ImageNet squeezing setting.

| config | value |
|---|---|
| optimizer | AdamW |
| base learning rate | 0.001 |
| weight decay | 0.01 |
| optimizer momentum | $\beta_1, \beta_2 = 0.9, 0.999$ |
| batch size | 1,024 |
| learning rate schedule | cosine decay |
| training epoch | 300 |
| augmentation | RandomResizedCrop |

(b) ImageNet-1K validation setting.

| config | value |
|---|---|
| $\alpha_{\text{BN}}$ | 1.0 |
| optimizer | Adam |
| base learning rate | 0.1 |
| optimizer momentum | $\beta_1, \beta_2 = 0.5, 0.9$ |
| batch size | 100 |
| learning rate schedule | cosine decay |
| recovering iteration | 1,000 |
| augmentation | RandomResizedCrop |

(c) Tiny-ImageNet recovering setting.

| config | value |
|---|---|
| $\alpha_{\text{BN}}$ | 0.01 |
| optimizer | Adam |
| base learning rate | 0.25 |
| optimizer momentum | $\beta_1, \beta_2 = 0.5, 0.9$ |
| batch size | 100 |
| learning rate schedule | cosine decay |
| recovering iteration | 2,000 |
| augmentation | RandomResizedCrop |

(d) ImageNet-1K recovering setting.

Table 7: Hyper-parameter settings in three stages.

• ImageNet-1K: We use PyTorch off-the-shelf ResNet-{18, 50} with *V1* weights and Top-1 accuracy of {69.76%, 76.13%} as squeezed/condensed models. In the original training script [39], ResNet models are trained for 90 epochs with a SGD optimizer, learning rate of 0.1, momentum of 0.9 and weight decay of $1 \times 10^{-4}$.

**BN-ViT Model Structure.** To give a clear illustration of incorporating BN layers into ViT, we provide more details in this subsection. The definitions of vanilla ViT and BN-ViT are presented below to show the structure modification. The vanilla ViT can be formulated as:

$$\mathbf{z}'_\ell = \text{MHSA}\left(\text{LN}\left(\mathbf{z}_{\ell-1}\right)\right) + \mathbf{z}_{\ell-1}$$

$$\mathbf{z}_\ell = \text{FFN}\left(\text{LN}\left(\mathbf{z}'_\ell\right)\right) + \mathbf{z}'_\ell$$

where $\mathbf{z}'_\ell$ is the intermediate representation before Feed-forward Network (FFN), and $\mathbf{z}_\ell$ is that after FFN and residual connection. FFN contains two linear layers with a GELU non-linearity in between them, i.e.,

$$\text{FFN}(\mathbf{z}'_\ell) = \left(\text{GELU}\left(\mathbf{z}'_\ell W^1_\ell + b^1_\ell\right)\right) W^2_\ell + b^2_\ell$$

The newly constructed BN-ViT is:

$$\mathbf{z}'_\ell = \text{MHSA}\left(\text{BN}\left(\mathbf{z}_{\ell-1}\right)\right) + \mathbf{z}_{\ell-1}$$

$$\mathbf{z}_\ell = \text{FFN}_{\text{BN}}\left(\text{BN}\left(\mathbf{z}'_\ell\right)\right) + \mathbf{z}'_\ell$$

where we add one additional BN layer in-between two linear layers of FFN, i.e.,

$$\text{FFN}_{\text{BN}}(\mathbf{z}'_\ell) = \left(\text{GELU}\left(\text{BN}\left(\mathbf{z}'_\ell W^1_\ell + b^1_\ell\right)\right)\right) W^2_\ell + b^2_\ell$$

We follow the DeiT official training recipe to train a DeiT-Tiny-BN model for 300 epochs with an AdamW optimizer, cosine decayed learning rate of $5 \times 10^{-4}$, weight decay of 0.05 and 5 warmup epochs.

| Ablation | | | Top-1 acc. (%) | |
|---|---|---|---|---|
| $\mathcal{R}_{TV}$ | $\mathcal{R}_{\ell_2}$ | Random Crop | Tiny-ImageNet | ImageNet-1K |
| ✓ | ✓ | ✗ | 29.87 | 22.92 |
| ✓ | ✗ | ✗ | 29.92 | 23.15 |
| ✗ | ✓ | ✗ | 30.11 | 40.81 |
| ✗ | ✗ | ✗ | 30.30 | 40.37 |
| ✗ | ✗ | ✓ | 37.88 | 46.71 |

Table 8: Top-1 validation accuracy under regularization ablation settings. ResNet-18 is used in all three stages with the relabeling temperature $\tau = 20$.

### A.3 Recovering Details

**Regularization Terms.** We conduct a large number of ablation experiments under varying regularization term conditions, as illustrated in Table 8. The two image prior regularizers, $\ell_2$ regularization and total variation (TV), are not anticipated to enhance validation accuracy as our primary focus is on information recovery rather than image smoothness. Consequently, we exclude these two regularization terms from our experiments.

**Memory Consumption and Computational Cost.** Regarding memory utilization, the memory accommodates a pre-trained model, reconstructed data, and the corresponding computational graph during the data recovery phase. Unlike the MTT approach, which necessitates all model states across all epochs during training to align with the trajectory, our proposed methodology, SRe²L, merely requires the statistical data from each BN layer, stored within the condensed model, for synthetic image optimization. In terms of total computational overhead, it is directly proportional to the number of recovery iterations. To establish a trade-off between performance and computational time, we enforce a recovery budget of 1k iterations for Tiny-ImageNet and 2k iterations for ImageNet-1K in ablation experiments. Our best accuracy, achieved on condensed data from 4k recovery iterations, is presented in Table 4 in the main paper.

**Hyper-parameter Setting.** We calculate the total recovery loss $\ell_{total} = \underset{\mathcal{C}_{\text{syn}},|\mathcal{C}|}{\arg\min} \ell\left(\phi_{\boldsymbol{\theta}_{\mathcal{T}}}(\widetilde{\boldsymbol{x}}_{\text{syn}}), \boldsymbol{y}\right) + \alpha_{\text{BN}}\mathcal{R}_{\text{BN}}$ and update synthetic data with the parameter setting in Table 7c and Table 7d for Tiny-ImageNet and ImageNet-1K, respectively.

### A.4 Relabeling & Validation Details

In this experiment, we utilize an architecture identical to that of a recovery model to provide soft labels as a teacher for synthesized images. We implement a fast knowledge distillation process to isolate the utilization of teacher models in post validation training with a training budget of 300 epochs and a temperature setting of $\tau = 20$.

**Hyper-parameter Setting.** Regarding Tiny-ImageNet, we leverage the condensed data and the retargeted labels to train the validation model over a span of 100 epochs, with all other training parameters adhering to the condensing configurations outlined in Table 7a. In the case of ImageNet-1K, we apply CutMix augmentation with mix probability $p = 1.0$ and Beta distribution $\beta = 1.0$ to train the validation model in accordance with the parameter configurations presented in Table 7b.

## B  Low-Resolution Data

To demonstrate our method's effectiveness on low-resolution datasets, we first conduct additional ablation experiments on ImageNet-1K dataset, including down-sampled resolutions of $112 \times 112$ and $64 \times 64$. In Table 9, the validation accuracy increases as the image resolution size grows, demonstrating that our method is more suitable for handling high-resolution datasets with better effectiveness.

We then conduct experiments on the CIFAR datasets to demonstrate the efficacy of our method when applied to small-scale datasets with fewer classes and a lower image resolution of $32 \times 32$. The adapted ResNet-18 is used as a backbone model throughout our SRe²L's three phases, and the hyper-parameter settings are presented in Table 10. As shown in Table 11, our results on the relatively

| Dataset (IPC=50) | ResNet-18 | ResNet-50 | ResNet-101 |
|---|---|---|---|
| IN-1K-64×64 | 35.27 | 42.26 | 44.37 |
| IN-1K-112×112 | 34.15 | 42.76 | 45.25 |
| IN-1K-224×224 | 46.75 | 55.62 | 60.81 |

Table 9: Top-1 validation accuracy on distilled and down-sampled ImageNet-1K datasets.

large CIFAR-100 dataset are on par with those of leading-edge methods, such as DM, FrePo, MTT, and TESLA, among others. However, on the smaller CIFAR-10 dataset, a clear gap exists. Overall, the additional CIFAR experiments suggest that our approach might also offer significant benefits for the lower-resolution dataset, and we highlight that our approach continues to demonstrate superior computational efficiency and enhanced processing speed when applied to these datasets.

| config | value |
|---|---|
| optimizer | SGD |
| base learning rate | 0.1 |
| momentum | 0.9 |
| weight decay | 5e-4 |
| batch size | 128 |
| learning rate schedule | cosine decay |
| training epoch | 200 (squeeze) / 400 (val) |
| augmentation | RandomCrop |

(a) Squeezing/validation setting.

| config | value |
|---|---|
| $\alpha_{BN}$ | 0.01 |
| optimizer | Adam |
| base learning rate | 0.25 |
| optimizer momentum | $\beta_1, \beta_2 = 0.5, 0.9$ |
| batch size | 100 |
| temperature | 30 |
| learning rate schedule | cosine decay |
| recovery iteration | 1,000 |

(b) Recovery setting.

Table 10: Hyper-parameter settings on CIFAR-100.

| IPC | DM [10] | FrePo [5] | MTT [1] | TESLA [11] | SRe$^2$L |
|---|---|---|---|---|---|
| 10 | $29.70 \pm 0.30$ | $41.30 \pm 0.20$ | $40.10 \pm 0.40$ | $\mathbf{41.70 \pm 0.30}$ | $23.48 \pm 0.80$ |
| 50 | $43.60 \pm 0.40$ | $44.30 \pm 0.20$ | $47.70 \pm 0.20$ | $47.90 \pm 0.30$ | $\mathbf{51.35 \pm 0.79}$ |
| 100 | – | – | – | – | $\mathbf{57.06 \pm 1.23}$ |
| 200 | – | – | – | – | $\mathbf{59.64 \pm 1.24}$ |

Table 11: Top-1 validation accuracy on distilled CIFAR-100 dataset.

## C   Feature Embedding Distribution

We feed the image data through a pretrained ResNet-18 model, subsequently extracting the feature embedding prior to the classification layer for the purpose of executing t-SNE [40] dimensionality reduction and visualization. Fig. 7a exhibits two distinct feature embedding distributions of synthetic Tiny-ImageNet data, sourced from 3 classes in MTT's and SRe$^2$L's condensed datasets, respectively. Relative to the distribution present in MTT, SRe$^2$L's synthetic data from differing classes displays a more dispersed pattern, whilst data from identical classes demonstrates a higher degree of clustering. This suggests that the data synthesized by SRe$^2$L boasts superior discriminability with respect to feature embedding distribution and can therefore be utilized to train models to attain superior performance. Fig. 7b illustrates feature embedding distributions of SRe$^2$L's synthetic ImageNet-1K data derived from 8 classes. Our synthetic ImageNet-1K data also exemplifies exceptional clustering and discriminability attributes.

## D   Theoretical Analysis

We provide a theoretical analysis of the generalization ability on the condensed dataset. Dataset condensation task generally aims to train on the condensed data meanwhile achieving good performance on the original val data. Given the significance of estimating the generalization error (GE) of deep neural networks as a method for evaluating their ability to generalize, we adopt this approach for

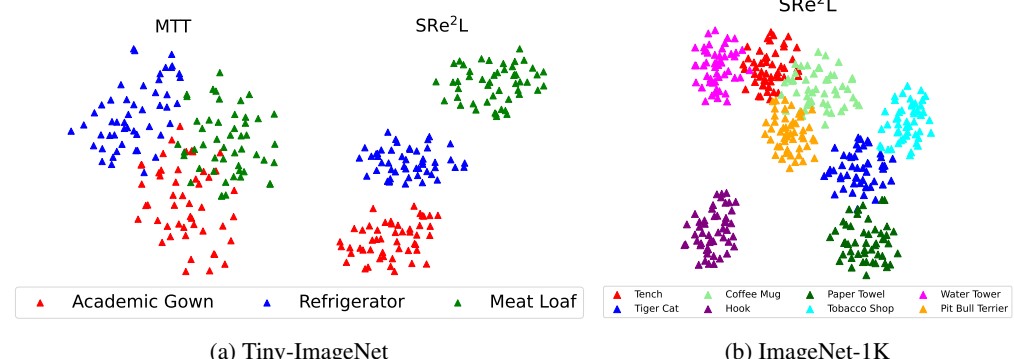

(a) Tiny-ImageNet                    (b) ImageNet-1K

Figure 7: Feature embedding distribution on synthetic data of Tiny-ImageNet and ImageNet-1K. ResNet-18 is used as the feature embedding extractor.

assessing the generalization capability of our condensed dataset for analyzing the generalization/error bounds between models trained on original data and condensed data.

Specifically, we employ the Mutual Information (MI) between the original/condensed input and the final layer representations to carry out this analysis, using the same network architecture limit to bound MI, in line with the methodology outlined in [41]. To elaborate further:

The MI between two variables $X$ and $D$ is:

$$I(X; D) \equiv \sum_{x,d} p(x,d) \log \frac{p(x,d)}{p(x)p(d)} = \mathbb{E}_{p(x,d)} \left[ \log \frac{p(d \mid x)}{p(d)} \right]$$

where $X$ is the input sample, $D$ is the input representation, i.e., model's output. The *leave one out* upper bound (UB) [42] can be utilized to conservatively bound MI:

$$I(X; D) \leq \mathbb{E} \left[ \frac{1}{N} \sum_{i=1}^{N} \log \frac{p(d_i \mid x_i)}{\frac{1}{N-1} \sum_{j \neq i} p(d_i \mid x_j)} \right] = I_{\mathrm{UB}}$$

Following the information theory fundamentals, by applying the conventional Probably Approximately Correct (PAC) of GE bound, we can obtain the bound on GE as:

$$\mathrm{GE} < \sqrt{\frac{\log(|\mathcal{H}|) + \log(1/\delta)}{2N_{\mathrm{trn}}}}$$

where $|\mathcal{H}|$ is the hypothesis-class cardinality and $N_{\mathrm{trn}}$ is the number of training examples. For the synthetic data, $N_{\mathrm{trn}} = |\mathcal{C}_{\mathrm{syn}}|$, while for the full data, $N_{\mathrm{trn}} = N_{\mathrm{ori}}$. The confidence parameter, denoted by $\delta$ and ranging between 0 and 1, specifies the likelihood that the bound remains consistent with respect to the chosen $N_{\mathrm{trn}}$ training samples.

According to the property of deep neural networks [41], the cardinality of the hypothesis space reduces to $|\mathcal{H}| \approx 2^{|\mathcal{T}|}$ where $|\mathcal{H}|$ is the number of class-homogeneous clusters that the backbone network distinguishes. An estimate for the number of clusters can then be obtained by $|\mathcal{T}| \approx 2^{H(X)}/2^{H(X|Z)} = 2^{I(X;Z)}$.

The ability of Input Compression Bound (ICB) [43, 44] is to predict changes in GE under different dataset interventions, it then can be formulated as:

$$\mathrm{GE}_{\mathrm{ICB}} < \sqrt{\frac{2^{I(X;D)} + \log(1/\delta)}{2N_{\mathrm{trn}}}}$$

Thus, we can have the generalization error bound for the condensed data as:

$$\mathrm{GE}_{\mathrm{ICB}}^{\mathrm{syn}} < \sqrt{\frac{2^{I(X;D)} + \log(1/\delta_{\mathrm{syn}})}{2|\mathcal{C}_{\mathrm{syn}}|}}$$

where the generalization error bound for full dataset is $\mathrm{GE}_{\mathrm{ICB}}^{\mathrm{full}} < \sqrt{\frac{2^{I(X;D)} + \log(1/\delta_{\mathrm{full}})}{2N_{\mathrm{trn}}}}$.

# E  More Visualization of Synthetic Data

We provide more visualization comparisons on synthetic Tiny-ImageNet between MTT and $\text{SRe}^2\text{L}$ in Fig. 8. Additionally, we visualize synthetic samples pertaining to ImageNet-1K in Fig. 9 and Fig. 10 for a more comprehensive understanding. It can be observed that our synthetic data has the stronger semantic information than MTT with more object textures, shapes and details, which demonstrates the superior quality of our synthesized data.

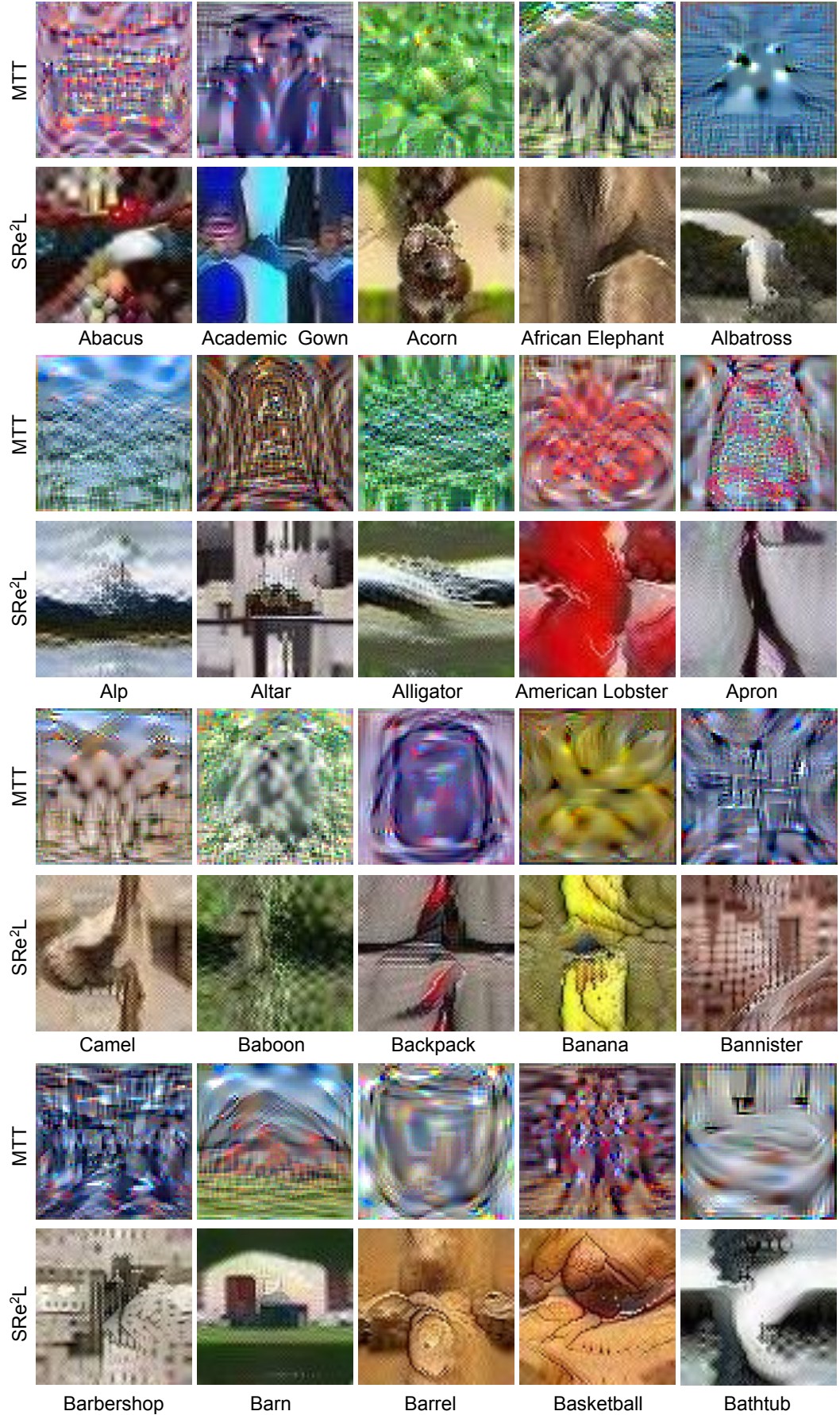

Figure 8: Synthetic data visualization on Tiny-ImageNet from MTT [1] and SRe$^2$L.

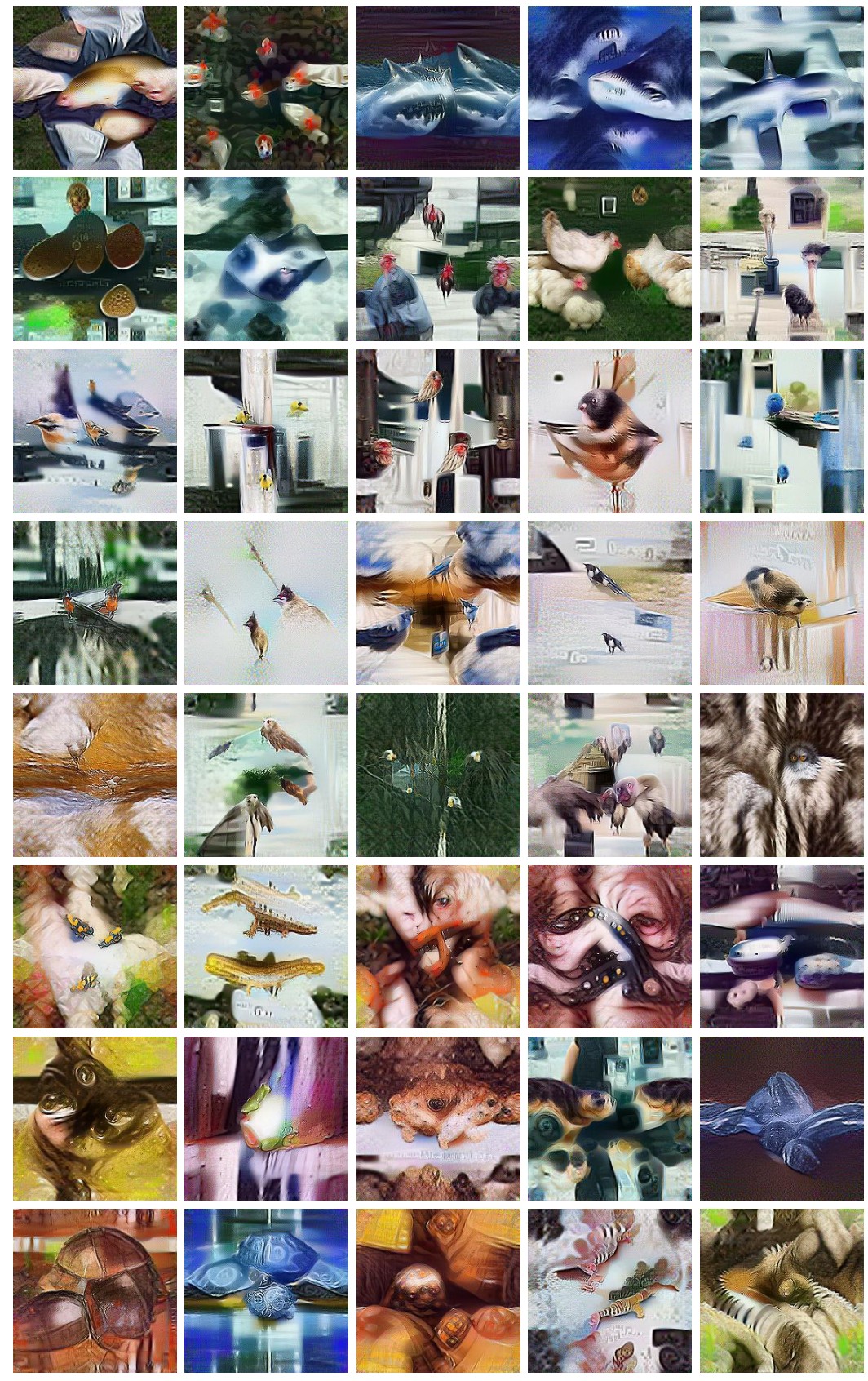

Figure 9: Synthetic data visualization on ImageNet-1K from SRe$^2$L.

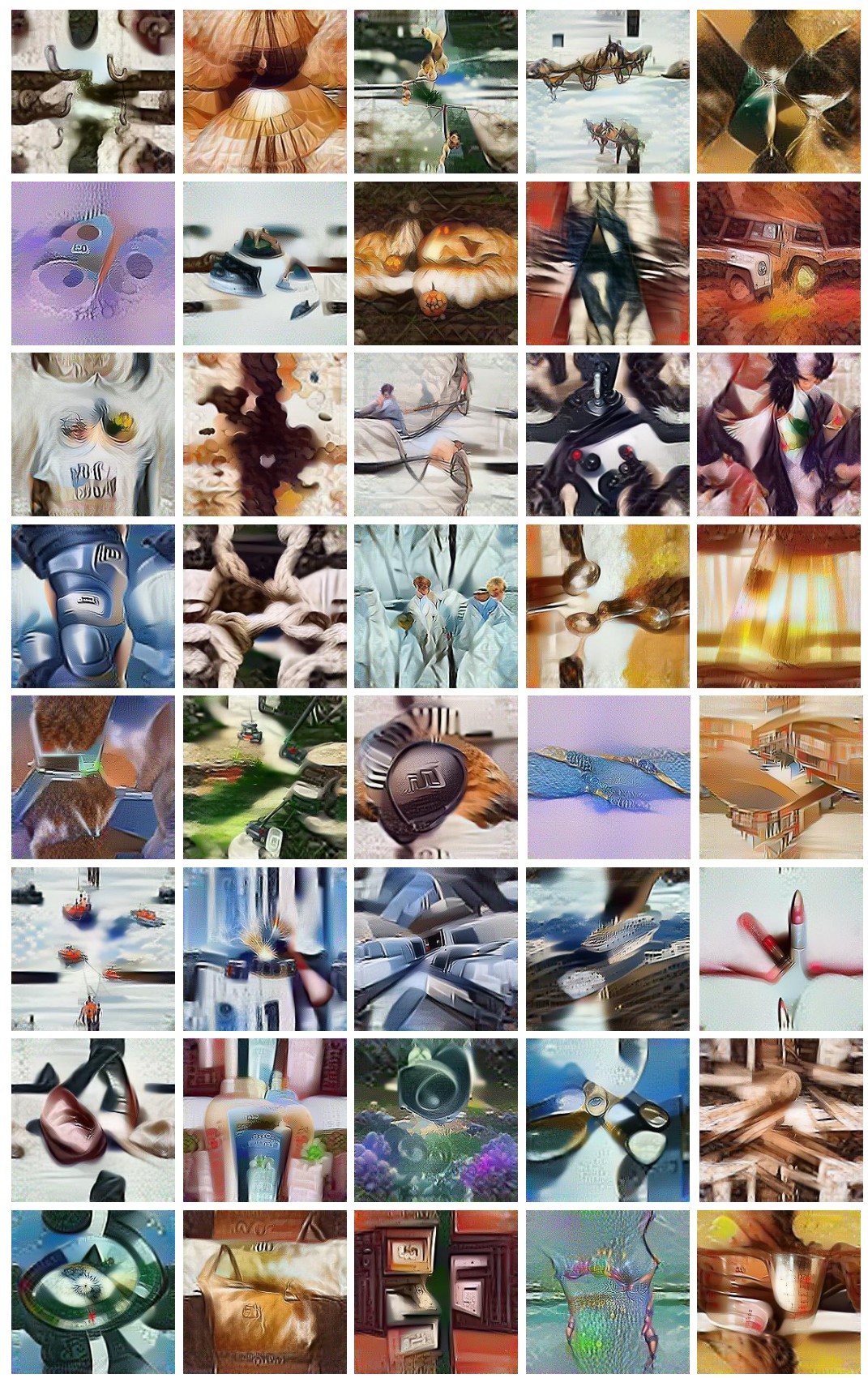

Figure 10: Synthetic data visualization on ImageNet-1K from SRe$^2$L.

