# OpenReview forum: "Squeeze, Recover and Relabel: Dataset Condensation at ImageNet Scale From A New Perspective"
_NeurIPS.cc/2023/Conference — NeurIPS 2023 spotlight_

### Official Review · Reviewer_7q3V · 2023-06-28

**Soundness:** 3 good
**Presentation:** 3 good
**Contribution:** 3 good
**Rating:** 6
**Confidence:** 4

**Summary:**

The paper addresses the dataset condensation task and proposes a new framework termed Squeeze, Recover and Relabel. In this three step approach, the authors first train a model from scratch to accommodate most of the crucial information from the original dataset. In the second stage, target data is synthesized from Gaussian noise. And in the final stage, the generated synthetic data is relabelled using a crop-level scheme to align with the true label of the data. Extensive and controlled experimentation showed significant performance improvement compared to previous state-of-the-art methods.

**Strengths:**

This publication has several strengths including:

1) The writing is clear and easy to understand.

2) Generalizability of the framework to scale of datasets, input resolution, and the size of network architectures.

3) Good experimental methodology with carefully designed ablations that justifies architectural design decisions especially impacts of  squeezing budget, diverse data augmentations for original data compression, recovery budget and regularization terms for data recovery and insights on model choice and training for relabeling process.

4) Very exhaustive in-depth empirical comparison with state-of-the-art methods demonstrating strong performance as well as incur reduced compute and memory consumption.

**Weaknesses:**

I am confused by the claims in the section “Cross-Architecture Generalization”. The authors attribute the suboptimal performance of DeiT-Tiny on the condensed datasets due to the model’s inherent need for substantial training data.

However, from Table 5 there appears to be a “cross-architecture gap”: ResNet based evaluation models perform poorly on condensed data based on ViT based squeezed model compared to when evaluated on condensed data based on ResNet based squeezed model. Similar observation for DeiT-Tiny evaluated on condensed data based on DeiT-Tiny-BN and ResNet-18 squeezed model. This shows that the condensed does not generalize well to different network architectures.

**Questions:**

The paper in its current form needs clarification on Cross-Architecture Generalization. Please refer to the “Weakness” section for details.

The authors acknowledge the performance disparity between condensed dataset in the limitation section. I feel that this might limit the adaptability of the proposed approach.

Minor typo:
1) Line 277 refers to Figure 3, it should refer to Table 5.
2) “Deit” -> “DeiT”

**Limitations:**

Yes, the authors discuss the limitations in the paper.

---

> ### Author Rebuttal · Authors · 2023-08-09
>
> We sincerely thank you for your constructive comments. We are encouraged that you find our work clear and easy to understand, and provide an exhaustive in-depth empirical comparison with state-of-the-art methods. We would like to address the comments and questions below.
>
> >Q1. Clarification on Cross-Architecture Generalization.
>
> Thanks for the insightful suggestion. It is a well-known observation in prior dataset condensation/distillation works that performance degradation often results from a mismatch between the synthesis and final training architectures. Yet, as indicated in the following results, the gap between ViT and ResNet-18 in our case is significantly narrower compared to the previous method TESLA. This suggests that our approach will be better at managing this mismatch with stronger *Cross-Architecture Generalization* ability.
>
> |       | ViT  | ResNet-18 |        Gap        |
> |:----- |:----:|:---------:|:-------------|
> | TESLA | 11.0 |    7.7    | $\downarrow$ 30.0%  |
> | Ours  | 25.4 |   24.7    | $\downarrow$ 2.8% |
>
> Moreover, Figure 3 clearly exhibits the proficiency of cross-model generalization across ResNet-{18, 50, 101}, and Table 5 presents cross-architecture generalization between ViT and ResNet. We will make this clearer in our revision.
>
> >Q2. The authors acknowledge the performance disparity between condensed dataset in the limitation section. I feel that this might limit the adaptability of the proposed approach.
>
> Thanks for your comments, we clarify the *adaptability* ability of our method manifests in at least two aspects:
>
> 1. The proposed method is the only approach capable of distilling the entire ImageNet-1K under 224 $\times$ 224 resolution, while still achieving commendable performance (60.8%) on the original ImageNet-1K validation set, which is 32.9% higher than the previous SOTA method TESLA (ICML 2023) using the same IPC of training samples. This has been shown the strongest *adaptability* ability in the task.
>
> 2. Moreover, our compression rate is 25$\times$. This implies that we are training our model on just 1/25 of the samples compared to conventional model training. This will be a good fit/adaptability for various resource-constrained training scenarios. Given this drastic reduction in training samples, the performance gap in comparison to full-data model training is surprisingly narrow.
>
> >Minor typo: 1. Line 277 refers to Figure 3, it should refer to Table 5.  2. “Deit” -> “DeiT”
>
> Thanks for pointing them out. Line 277 refers to both Figure 3 and Table 5 in our paper. We have corrected the typos in our revision and will polish the whole paper thoroughly.

---

> > ### Comment · Reviewer_7q3V · 2023-08-20
> >
> > I went over the rebuttal and the other reviews. I appreciate the authors addressing my raised concerns about the "Cross-Architecture Generalization” and  providing clarification on the adaptability ability. I suggest the authors add the above results and discussion to the revised paper. I am happy to increase my rating.

---

> > > ### Author Response · Authors · 2023-08-21
> > >
> > > Thank you for your valuable feedback. We will incorporate the results and discussions from the rebuttal into our revised paper. Best wishes to you.

---

### Official Review · Reviewer_MBn1 · 2023-07-02

**Soundness:** 3 good
**Presentation:** 3 good
**Contribution:** 2 fair
**Rating:** 5
**Confidence:** 3

**Summary:**

This paper proposes a 3-step dataset condensation approach. Instead of applying bilevel optimization based approach in the previous work, the proposed method break down 3 decoupled steps: squeeze, recover and relabel. The key idea is decoupling the modeling training on real data and the generation of the synthetic data. At the first squeeze step, the model is trained on the original full dataset. At the second recovery step, synthetic data is generated using pretrained model and class prior with additional regularization (TV loss and BN consistency). At the 3rd stage, soft labels are generated using the pretrained model. And finally, a model on synthetic data is trained using the images from step 2 and labels from step 3.

**Strengths:**

Compared with previous method on Tiny-IN & IN-1K, the proposed approach achieves superior performance.
The proposed method generates more visually appealing images from the example images in Fig 4.
The first work condenses the full IN-1K, based on the claim from the paper.
Efficient way of using decoupled steps during the generation.

**Weaknesses:**

1. Lack of strong technical novelty. The proposed method combines a few prior work on image synthesis, such as deepDream & Inverting Image, and applies it directly into a new problem.
2. Lack of comprehensive study with baseline approaches, and demonstration of the improvement of the key novelty.

**Questions:**

I am not familiar with the field, but I am not convinced about the overall problem being solved.
If we would like to condense the original dataset for efficient training, we should at least achieve comparable accuracy on retraining. This is not the case from the IN-1K result.
If we would like to obtain a way of retraining a new model without requiring original labeled dataset, semi-supervised learning on a unlabeled data would be a more promising direction instead of image synthesis.

**Limitations:**

Please see my concerns in the Questions section

---

> ### Author Rebuttal · Authors · 2023-08-09
>
> We sincerely thank you for your constructive comments. We are appreciated that you find our work achieving superior performance with more visually appealing images. We would like to address the comments and questions below.
>
> >W1. Strong technical novelty. The proposed method combines a few prior work on image synthesis, such as deepDream & Inverting Image, and applies it directly into a new problem.
>
> Thanks for the valuable feedback. We highlight our novelty in each stage of *squeeze*, *recover* and *relabel* comparing to prior methods such as DeepDream, Inverting Image, etc.
>
> 1. In the squeeze phase, we have crafted a new BN-based ViT architecture specifically for the task of data condensation. Interestingly, we've also observed that a higher-performing model in the squeeze phase does not necessarily translate to superior knowledge for the subsequent processes of recover and relabel. This is discovered by our conducted extensive experiments with various data augmentation techniques, which help to identify the most useful strategies that enhance the squeeze procedure.
> 2. In the recover phase, we performed systematic ablation studies on various regularizers to understand their impact on relabeling and final training. Additionally, we introduced a simple yet highly effective multi-crop optimization technique, which significantly elevated the performance levels.
> 3. In the relabel phase, we go beyond the straightforward application of FKD. We enhance it by developing a novel soft label storage mechanism. This innovative solution ensures FKD's compatibility with mixture-based data augmentation techniques like Mixup and CutMix, which is not supported in FKD's vanilla design.
>
> >W2. Comprehensive study with baseline approaches, and demonstration of the improvement of the key novelty.
>
> Thanks for the suggestion. In response, we have expanded our analysis to include additional comparisons with baseline approaches as detailed below:
>
> We recover images from ConvNet in $\texttt{SRe}^2\texttt{L}$, the IPC equals 10 in post-training for the comparison with baseline method TESLA. We conducted two configurations: relabel using the same squeezed model and relabel using the large pretrained ResNet-18. The results are shown in the following table and our approach demonstrates superior performance over the baseline. The results also reflect the observation and novelty in our paper that a correct relabeling model is necessary for the final accuracy. We will include these comparisons in our revised paper.
>
> | Squeezed Model      | ResNet-18 | ResNet-50 | ResNet-101 |
> | ----------------------------------- |:---------:|:---------:|:----------:|
> | TESLA   (IPC=10)                    |    7.7    |    --     |     --     |
> | Our ConvNet (relabel w/ itself, IPC=10)   |   17.0	|20.5	| 21.2 |
> | Our ConvNet (relabel w/ ResNet-18, IPC=10) |   12.8 |	15.7 | 17.1 |
> | Our ResNet-18 (IPC=10)                   |   21.3   |   28.4   |    30.9   |
>
> >Q1. I am not familiar with the field, but I am not convinced about the overall problem being solved. If we would like to condense the original dataset for efficient training, we should at least achieve comparable accuracy on retraining. This is not the case from the IN-1K result. If we would like to obtain a way of retraining a new model without requiring original labeled dataset, semi-supervised learning on a unlabeled data would be a more promising direction instead of image synthesis.
>
> Thank you for your insights. While our proposed method may not completely resolve the dataset condensation problem, it significantly advances the ability to address this challenge within this domain:
>
> 1. It is worth noting that ours is the only approach capable of distilling the entire ImageNet-1K under 224 $\times$ 224 resolution, while still achieving commendable performance (60.8%) on the original validation set. It is an absolute 32.9% higher than the previous SOTA method TESLA (ICML 2023) using the same IPC training samples.
>
> 2. Moreover, our compression rate is 25$\times$. This implies that we are training our model on just 1/25 of the samples compared to conventional model training. Given this drastic reduction in training samples, it's reasonable that the performance may not fully reach that of standard model training.
>
> We also clarify that our objective is not to pursue a method that eliminates the need of the original labeled dataset. Instead, our focus is on data condensation/distillation where the goal of this task is to achieve a similar level of accuracy as the original full dataset but with **much fewer samples**. This target makes data distillation work apart from semi-supervised learning, which has a different set of objectives.
>
> We appreciate the reviewer's valuable feedback. We will persist in our efforts to enhance and optimize the performance of data distillation within this area.

---

### Official Review · Reviewer_RJ8s · 2023-07-03

**Soundness:** 2 fair
**Presentation:** 2 fair
**Contribution:** 2 fair
**Rating:** 6
**Confidence:** 3

**Summary:**

This paper proposes a new dataset condensation termed Squeeze, Recover, and Relabel that decouples the bilevel optimization of model and synthetic data during training. Extensive experiments show the effectiveness and efficiency of the proposed method in several IPC settings.

**Strengths:**

1, The paper is well-written and easy to understand.

2, The proposed method becomes efficient due to the decoupled stages.

3, The extensive experiments show the effectiveness of the method.

**Weaknesses:**

1, Albeit the computation and memory efficiency in the proposed method, the whole training time might be comparable to other methods like DM and MTT.

2, Unclear illustration about BN layers in VIT. Is it means that the proposed method uses the BN layer to replace the LN?

3, The lack of comparison between the IPC and directly sampling the same scale images.

**Questions:**

1, One naive question: why the condensed images could be used for training? I am not familiar with this topic.

2, The stage-3 Relabel seems like the process of knowledge distillation. Could the author try another setting, like using R50 to teach R18?

---

> ### Author Rebuttal · Authors · 2023-08-09
>
> We sincerely thank you for your constructive comments. We are encouraged that you find our work well-written and easy to understand, the proposed method is efficient with extensive experiments. We would like to address the comments and questions below.
>
> >W1. Albeit the computation and memory efficiency, the whole training time might be comparable to DM and MTT.
>
> Thanks for pointing out this. We've broken down the whole training time consumption at each stage of our training process in the table below. The timing is evaluated on Tiny-ImageNet using the *ConvNet/ResNet-18* architectures with one RTX 4090 GPU.
>
> |  | Squeeze/Pretrain (h)| Recover/Synthesis (h)| Relabel/Distilling (h)| Total (h)|
> |:----- |:-------:|:------:|:-----:|:-----:|
> | DM| - | 23.78/99.88 |-| 23.78/99.88 |
> | MTT| 0.15/0.63  $\times$ 100 |16.60/69.72| - | 31.60/132.72 |
> | TESLA | 0.15/0.63  $\times$ 100 |16.93/71.11| 0.02/0.08| 31.95/134.19 |
> | Ours|0.37/1.54|1.98/8.33 |0.02/0.08|2.37/9.95|
>
> From this, it becomes evident that, even when considering the time spent on the squeeze/pre-training stage, our proposed framework still significantly surpasses the efficiency of other methods, such as DM and MTT.
>
> In the squeeze phase, multiple squeezed models are necessary to be pretrained for sampling and matching multiple trajectories in MTT. However, only one squeezed model is required to match Batch Norm statistics in our method. Consequently, the squeezing phase in our method is considerably more time-efficient compared to that of MTT.
>
> In the synthesis phase, we have included a time consumption comparison in Table 1 of our paper. This comparison illustrates the time taken to generate one Tiny-ImageNet image with a single iteration update. When considering the iteration settings for synthesis, both DM and MTT models train over 10K iterations. In contrast, our model requires only 4K iterations. Therefore, our model's reduced per-iteration synthesis times, resulting in a shorter total synthesis time.
>
> >W2. Unclear illustration about BN layers in VIT.
>
> The vanilla ViT can be formulated as:
>
> $$\mathbf z_{\ell}^{\prime}=\operatorname{MHSA}\left(\mathrm{LN}\left(\mathbf z_{\ell-1}\right)\right)+\mathbf z_{\ell-1}$$
>
> $$\mathbf z_{\ell}=\operatorname{FFN}\left(\mathrm{LN}\left(\mathbf z_{\ell}^{\prime}\right)\right)+\mathbf z_{\ell}^{\prime}$$
>
> where $\mathbf z_{\ell}^{\prime}$ is the intermediate representation before Feed-forward Network ($\operatorname{FFN}$), and $\mathbf z_{\ell}$ is that after $\operatorname{FFN}$ and residual connection. $\operatorname{FFN}$ contains two linear layers with a GELU non-linearity in between them, i.e.,
>
> $$\operatorname{FFN}(\mathbf z_{\ell}^{\prime})=\left(\operatorname{GELU}\left(\mathbf z_{\ell}^{\prime} W^1_\ell+b^1_\ell\right)\right) W_\ell^2+b_\ell^2$$
>
> The newly constructed BN-ViT is:
>
> $$\mathbf z_{\ell}^{\prime}=\operatorname{MHSA}\left(\mathrm{BN}\left(\mathbf z_{\ell-1}\right)\right)+\mathbf z_{\ell-1}$$
>
> $$ \mathbf z_{\ell}=\operatorname{FFN_{BN}}\left(\mathrm{BN}\left(\mathbf z_{\ell}^{\prime}\right)\right)+\mathbf z_{\ell}^{\prime}$$
>
> where we add one additional BN layer in-between two linear layers of $\operatorname{FFN}$, i.e.,
>
> $$
> \operatorname{FFN_{BN}}(\mathbf z_{\ell}^{\prime})=\left(\operatorname{GELU}\left(\operatorname{BN}\left(\mathbf z_{\ell}^{\prime} W^1_\ell+b^1_\ell\right)\right)\right) W_\ell^2+b_\ell^2
> $$
>
> We will include these details in our revision.
>
> >W3. Comparison between the IPC and directly sampling the same scale images.
>
> The results of directly sampling the same scale images are shown in the following table. As suggested by the reviewer, we randomly sampled 50 images per class as the pruned dataset. It can be observed that our dataset condensation results outperform the dataset pruning by large margins across various architectures.
>
> |  IPC=50  | ResNet-18 | ResNet-50 | ResNet-101 |
> |:--------|:--------:|:--------:|:---------:|
> | Random Dataset Pruning|27.47| 26.05 | 26.96 |
> | Dataset Condensation (Ours)|46.75| 55.62| 60.81|
>
> >Q1. One naive question: why the condensed images could be used for training? I am not familiar with this topic.
>
> We understand this concern stems from the distinctive visual appearance of synthetic data. Despite the condensed images appearing quite different from the natural ones, they are engineered to encapsulate the essence or core characteristics of the original large dataset, particularly:
>
> 1. Information Retention: These distilled images retain the important features and patterns required for effective training. In other words, despite their smaller size, distilled images retain crucial information from the original dataset. This information is what the model learns to recognize and apply when it sees new, similar data.
>
> 2. Noise Reduction: Dataset condensation/distillation can also serve as a form of noise reduction, filtering out unnecessary or irrelevant information in the synthetic data and allowing the model to focus on the most salient features.
>
> Furthermore, the visualization of distilled examples in Figure 1 of our supplementary material, offers a straightforward interpretation. It reveals that numerous small areas, saturated with categorical features, are scattered throughout the image. This distribution significantly augments the image's expressiveness, enriching its visual representation during model training.
>
> >Q2. The stage-3 Relabel seems like the process of knowledge distillation. Could the author try another setting, like using R50 to teach R18?
>
> Thanks for the suggestion. Indeed, we have accommodated a variety of results with diverse relabeling settings including the suggested *using R50 to teach R1* in Figure 3 of the paper. We have also included detailed configurations within the legends of each subfigure. For example, the notation $\mathrm{T_{R50}S_{R18}}$ is used to indicate the use of a ResNet-50 model for relabeling synthetic data, which subsequently teaches the learning process of a ResNet-18 model.

---

> > ### Comment · Reviewer_RJ8s · 2023-08-21
> >
> > The authors's rebutal well addressed my concerns. Thanks for the authors' efforts. I would like to raise my rate.

---

### Official Review · Reviewer_QGHe · 2023-07-04

**Soundness:** 4 excellent
**Presentation:** 4 excellent
**Contribution:** 4 excellent
**Rating:** 8
**Confidence:** 5

**Summary:**

This paper proposes a dataset distillation or dataset condensation method that can support ImageNet-scale compression. The main idea is inspired by some data-free knowledge distillation techniques to optimize the cross-entropy error, BN statistic distance, and some other prior terms for the distilled data. The complexity is simple compared with recent mainstream methods of dataset distillation. And the proposed method achieves impressive accuracy for models trained with distilled data on large-scale datasets.

**Strengths:**

1. The proposed method is simple yet effective, which enjoys satisfactory scalability.
2. It is good to know for the community that dataset distillation could achieve promising results on large-scale datasets like ImageNet1k-224 resolution.
3. The writing is coherent and it's easy for readers to follow the proposals.

**Weaknesses:**

1. I am a little bit worried about the technical novelty. Since the main idea is largely inspired by data-free knowledge distillation techniques [a], I list this as a weakness for NeurIPS, the top-tier machine learning conference.
2. Some detailed ablation studies are expected:
    * I notice that the settings of this paper is different from previous works. For example, previous works typically use Convent for dataset distillation while this work mainly considers ResNet. I am not arguing that the setting must be the same. However, I do think that an ablation study is necessary to show the improvement coming from the architectures.
    * A sensitivity analysis with respect to $\alpha_{BN}$ is necessary.
    * I do understand that the proposed method is mainly for large-scale datasets. But it is also interesting to dynamically increase the size of datasets and compare the performance with the existing methods, to help readers better understand when the proposed method yields advantages.

[a] Dreaming to distill: Data-free knowledge transfer via deepinversion. Yin et al., CVPR 2020.

**Questions:**

My major questions focus on the detailed ablation studies. Please refer to the 2nd point of the above weaknesses part for details.

**Limitations:**

The authors should clarify more limitations of the specific solution they proposed instead of those general issues of dataset distillation, e.g., larger datasets and performance gap with models trained on full data.

---

> ### Author Rebuttal · Authors · 2023-08-09
>
> We sincerely thank you for your constructive comments. We are encouraged that you find our work simple yet effective,  enjoying satisfactory scalability, and helping the community to know dataset distillation could achieve promising results on large-scale datasets. We would like to address the comments and questions below.
>
> > W1. The technical novelty.
>
> Thanks for your kind comments. We highlight our novel contributions in this work beyond data-free knowledge distillation and other related works such as DeepDream, Inverting Image, etc., in the following aspects:
>
> 1. In the squeeze phase, we have crafted a new BN-based ViT architecture specifically for the task of data condensation. Interestingly, we've also observed that a higher-performing model in the squeeze phase does not necessarily translate to superior knowledge for the subsequent processes of recover and relabel. This is discovered by our conducted extensive experiments with various data augmentation techniques, which help to identify the most useful strategies that enhance the squeeze procedure.
> 2. In the recover phase, we propose the new perspective that not all regularizers in image synthetic are useful for dataset condensation. We performed systematic ablation studies on various regularizers to understand their impact on relabeling and final training. Additionally, we introduced a simple yet highly effective multi-crop optimization technique, which significantly elevated the performance levels.
> 3. In the relabel phase, we go beyond the straightforward application of FKD. We enhance it by developing a novel soft label storage mechanism. This innovative solution ensures FKD's compatibility with mixture-based data augmentation techniques like Mixup and CutMix, which is not supported in FKD's vanilla design.
>
> > W2 (1). Previous works typically use Convent for dataset distillation while this work mainly considers ResNet. I am not arguing that the setting must be the same. However, I do think that an ablation study is necessary to show the improvement coming from the architectures.
>
> Thanks for the valuable comments. As suggested, we've incorporated additional ablation experiments to reconstruct images using ConvNet, as shown in the table below. To integrate BN-matching within $\texttt{SRe2L}$, we've implemented BatchNorm operations post each convolutional layer. Subsequently, after training the ConvNet-BN model for 90 epochs, we achieved a well-optimized model boasting a Top-1 accuracy rate of 58.62% as its squeezed model. The results under IPC=10 across various post-training architectures still outperform TESLA by large margins,  highlighting $\texttt{SRe2L}$'s proficiency even with the non-residual model.
>
> | Squeezed Model  (IPC=10)  | ResNet-18 | ResNet-50 | ResNet-101 |
> | ------- |:---------:|:---------:|:----------:|
> | TESLA  | 7.7  |  -  |  -  |
> | ResNet-18 (paper)  |  21.3 | 28.4 | 30.9 |
> | Our ConvNet  (relabel w/ itself) | 17.0 | 20.5 | 21.2 |
> | Our ConvNet (relabel w/ ResNet-18) | 12.8 | 15.7| 17.1 |
>
> > W2 (2). A sensitivity analysis with respect to $\alpha_{BN}$ is necessary.
>
> The experimental ablation results of $\alpha_{BN}$ are presented in the table below. $\alpha_{BN}=0.01$ achieves the highest accuracy on ResNet-101, which is adopted in our paper. While, $\alpha_{BN}=0.1$ obtains slightly better performance on ResNet-18 and ResNet-50 architectures. We will include these ablation results in our revised paper.
>
> | $\alpha_{BN}$ | ResNet-18 | ResNet-50 | ResNet-101 |
> | ------------- |:---------:|:---------:|:----------:|
> | 0.001         |   45.87   |   54.92   |   56.95    |
> | 0.01          |  46.75   |   55.62   | **60.81**  |
> | 0.1           | **47.83** | **56.19** |   58.36    |
> | 1.0           |   46.87   |   55.24   |   57.70    |
>
> > W2 (3). I do understand that the proposed method is mainly for large-scale datasets. But it is also interesting to dynamically increase the size of datasets and compare the performance with the existing methods.
>
> Thank you for the insightful suggestion. As recommended, we undertook further experiments from two dimensions: increasing resolution and increasing the number of classes. The details are presented in the table below.
>
> For the distillation of the *IN-1K-64x64* dataset, we adhered to the ResNet architecture previously employed for Tiny-ImageNet, adjusting the output dimension to accommodate 1K classes. When distilling the *IN-1K-112x112* dataset, we utilized the aforementioned adapted model and activated the max-pool layer to suit the 112 $\times$ 112 resolution.
>
> For the *IN-100-224x224* and *IN-10-224x224* datasets, we curated corresponding sub-datasets and aligned the standard ResNet's output dimension to match the respective class numbers. A trend emerged from our findings. As the number of classes decreased, accuracy exhibited an upward trajectory. This trend aligns seamlessly with established learning principles. The results from our experiments demonstrate that our approach is adaptable, catering not just to large datasets but also to datasets of diverse scales.
>
> | Dataset (IPC=50)| ResNet-18 | ResNet-50 | ResNet-101 |
> | ------- |:---------:|:---------:|:---------:|
> | IN-1K-224x224 (paper) | 46.75 |  55.62 | 60.81 |
> | IN-1K-112x112 | 34.15 | 42.76   | 45.25 |
> | IN-1K-64x64| 35.27 | 42.26 |   44.37    |
> |  |  |  |  |
> | IN-1K-224x224 (paper) | 46.75 |  - | - |
> | IN-100-224x224  | 52.70 | -  |  - |
> | IN-10-224x224  |  73.00 |  -  | -  |
>
> >L1. The authors should clarify more limitations of the specific solution they proposed instead of those general issues of dataset distillation.
>
> Thanks for the suggestion. Beyond the usual constraints associated with dataset distillation, we've noticed an additional limitation within our approach that the proposed framework requires storage of extra soft labels for the synthetic dataset, leading to increased disk storage consumption. In other aspects, our approach shows significant advantages over other methods.

---

> > ### Comment · Reviewer_QGHe · 2023-08-17
> > **Thanks for the rebuttal!**
> >
> > Thanks the author for the informative rebuttal! It clears most of my concerns. Here are some remaining ones:
> > 1. The authors have provided some results on down-sampled ImageNet datasets to illustrate the performance on small datasets, which conducts the ablation of dataset sizes via changing resolutions. I am curious about the studies with respect to changing the number of images and the number of classes. For example, how about the performance on small-scale datasets like CIFAR? The authors have mentioned in the supplement that the method is for large-scale datasets and the results on small datasets are not competitive. I definitely understand this. Nevertheless, I think it is important to report these results or some other forms of ablation studies on the number of images to help readers understand what sizes of datasets can benefit from this method.
> > 2. I notice that the authors use a trick of "Multi-crop Optimization", which is related to some recent works on synthetic data parameterization [a, b, c], and the performance is indeed sensitive to this operation. The authors are encouraged to have a discussion with these related works.
> >
> > [a] Dataset Condensation via Efficient Synthetic-Data Parameterization (Jang-Hyun Kim et al., ICML 2022)
> >
> > [b] Dataset Distillation via Factorization (Songhua Liu et al., NeurIPS 2022)
> >
> > [c] Remember the Past: Distilling Datasets into Addressable Memories for Neural Networks (Zhiwei Deng et al., NeurIPS 2022)

---

> > > ### Author Response · Authors · 2023-08-18
> > > **Further response to the comments**
> > >
> > > Thanks very much for the additional insightful comments and suggestions.
> > >
> > > >1. The authors have provided some results on down-sampled ImageNet datasets to illustrate the performance on small datasets, which conducts the ablation of dataset sizes via changing resolutions. I am curious about the studies with respect to changing the number of images and the number of classes. For example, how about the performance on small-scale datasets like CIFAR? The authors have mentioned in the supplement that the method is for large-scale datasets and the results on small datasets are not competitive. I definitely understand this. Nevertheless, I think it is important to report these results or some other forms of ablation studies on the number of images to help readers understand what sizes of datasets can benefit from this method.
> > >
> > > Adjusting the number of images can have a notable impact. Generally, having more images for each class in the original dataset tends to result in better-trained models. These improved models can often reproduce higher-quality images. As for altering the number of classes, we have presented the ImageNet results corresponding to varying class numbers, specifically, 1K, 100, and 10 classes, in the last three rows of the table in rebuttal of **W2 (3)**.
> > >
> > > Here, we incorporate additional ablation experiments on the CIFAR-10/100 datasets and the results are shown in the table below. The adapted ResNet-18 is utilized as a backbone model throughout our SRe2L's three phases. Our prior ablation findings showed that the proposed approach excels particularly with ImageNet scale, especially with resolutions exceeding 64 $\times$ 64 and classes numbering more than 100. On the relatively large CIFAR-100 dataset with more classes, our results parallel those of leading-edge methods, such as DM, CAFE, and MTT, among others. However, on the small CIFAR-10, the gap is clearly observed. Overall, the new CIFAR experiments suggest that our approach might not offer significant benefits for lower-resolution 32 $\times$ 32 datasets, such as CIFAR-10/100. We will include these results with discussions in our revised paper.
> > >
> > > | IPC | CIFAR-100 | CIFAR-10 |
> > > | --- |:---------:|:--------:|
> > > | 50  |   49.37   |  --   |
> > > | 100 |   54.08   |  60.97   |
> > > | 200 |   57.86   |  71.33   |
> > >
> > >
> > > >2. I notice that the authors use a trick of "Multi-crop Optimization", which is related to some recent works on synthetic data parameterization [a, b, c], and the performance is indeed sensitive to this operation. The authors are encouraged to have a discussion with these related works.
> > > [a] Dataset Condensation via Efficient Synthetic-Data Parameterization (Jang-Hyun Kim et al., ICML 2022)
> > > [b] Dataset Distillation via Factorization (Songhua Liu et al., NeurIPS 2022)
> > > [c] Remember the Past: Distilling Datasets into Addressable Memories for Neural Networks (Zhiwei Deng et al., NeurIPS 2022)
> > >
> > > Thanks for introducing these related papers. IDC [a] integrates multi-formation, updating various cropped regions in every iteration. When examined in our framework, the results indicated only a slight increase in accuracy, but this was offset by a considerably longer recovery time. Conversely, our approach utilizes a single-formation, updating just one cropped area at a time. This strategy is consistent with the RandomResizedCrop operation used during the post-training phase.
> > >
> > > [b] proposes a hallucinator-basis factorization method for the dataset distillation task. It leverages hallucinators to encode inner relations between different samples in original datasets, also introduces a pair of adversarial contrastive constraints to diversify the knowledge captured by different hallucinators. [b] argued that the data augmentations, e.g., multi-crop, cannot encode any information about the target datasets, and further proposes their approach to enhance the informativeness gained in distilled data.
> > >
> > > [c] proposes to learn a set of bases/memories which are shared between classes and combined through learned flexible addressing functions to generate a diverse set of training examples.
> > >
> > > We will definitely add the discussions with these related works in our revision.

---

> > > > ### Comment · Reviewer_QGHe · 2023-08-19
> > > >
> > > > Thanks the authors for the new results. Now, all my concerns are alleviated and I encourage the authors to include all these discussions in the final version. Currently, I find the contribution of this paper really strong for the community of dataset distillation, since large-scale datasets are of great challenge for existing methods and the authors have found a simple yet effective way to tackle the issue, which would be impactful for future works. I would like to increase my score to 8 - strong accept.

---

> > > > > ### Author Response · Authors · 2023-08-21
> > > > >
> > > > > Thank you very much for acknowledging the paper's value and contribution to the community. We'll include all the extra results and discussions in our updated paper. Have a great day!

---

### Official Review · Reviewer_4huB · 2023-07-05

**Soundness:** 3 good
**Presentation:** 3 good
**Contribution:** 4 excellent
**Rating:** 7
**Confidence:** 4

**Summary:**

The paper introduces a new dataset condensation framework, that is Squeeze, Recover, and Relabel (SRe2L).
SRe2L decouples the optimization of model and synthetic data during training, enabling effective condensation across varying dataset scales, model architectures, and image resolutions.

The authors mention in the paper about the advantages such as arbitrary resolution synthesis, low training cost and memory consumption, and scalability to different evaluation network architectures.

Extensive experiments on Tiny-ImageNet and full ImageNet-1K datasets demonstrate  its improved performance compared to state-of-the-art methods.
SRe2L also outperforms the MTT approach in terms of speed and memory consumption during data synthesis.
Overall, SRe2L presents a powerful solution for dataset condensation with improved performance and efficiency.

**Strengths:**

1. Clear Paper Organization: The paper exhibits a well-structured organization that aids in comprehending the presented concepts, methodologies, and experimental results. The logical flow of information allows readers to follow the paper's contributions easily.

2. Novel Framework: The authors propose a new dataset condensation framework, Squeeze, Recover, and Relabel (SRe2L), which offers a fresh perspective on addressing the data condensation problem. While some technical details may not be entirely novel, the paper presents an alternative solution to data condensation, introducing new ideas and approaches.

3. Surprising Performance: The scalability of data condensation to large datasets and deep networks poses significant challenges. The paper's performance on ImageNet-level datasets is particularly impressive, demonstrating the effectiveness and robustness of the proposed framework in handling complex datasets and achieving high validation accuracy.

**Weaknesses:**

1. Limited Theoretical Analysis: While the paper presents impressive empirical results and demonstrates the effectiveness of the proposed framework, it lacks a comprehensive theoretical analysis. More theoretical analysis, such as error bounds or performance upper bounds, would provide a deeper understanding of the underlying principles and limitations of the proposed approach. Incorporating theoretical analysis could further strengthen the paper's contributions and provide insights into the algorithm's behavior and performance guarantees.

Overall, the paper is well-structured, introduces a novel framework, and achieves remarkable performance on challenging datasets. However, enhancing the theoretical analysis would add a valuable dimension to the paper and provide a more comprehensive evaluation of the proposed method.

**Questions:**

see weakness

---

> ### Author Rebuttal · Authors · 2023-08-09
>
> We sincerely thank you for your constructive comments. We are encouraged that you find our work novel with surprising performance on scalability. We would like to address the comments and questions below.
>
> >W1. Limited Theoretical Analysis: While the paper presents impressive empirical results and demonstrates the effectiveness of the proposed framework, it lacks a comprehensive theoretical analysis. More theoretical analysis, such as error bounds or performance upper bounds, would provide a deeper understanding of the underlying principles and limitations of the proposed approach. Incorporating theoretical analysis could further strengthen the paper's contributions and provide insights into the algorithm's behavior and performance guarantees.
>
> Thanks for the insightful suggestion of incorporating a more theoretical perspective into our current analysis. Given the significance of estimating the generalization error (GE) of deep neural networks as a method for evaluating their ability to generalize, since the data condensation task aims to train on the condensed data meanwhile achieving good performance on the original val data. We've adopted this approach for assessing the generalization capability of our condensed dataset for analyzing the generalization/error bounds between models trained on original data and condensed data.
>
> More specifically, we employ the Mutual Information (MI) between the original/condensed input and the final layer representations to carry out this analysis, using the same network architecture limit to bound MI, in line with the methodology outlined in [1]. To elaborate further:
>
> The MI between two variables $X$ and $D$ is:
> $$I(X ; D) \equiv \sum_{x, d} p(x, d) \log \frac{p(x, d)}{p(x) p(d)}=\mathbb E_{p(x, d)}\left[\log \frac{p(d \mid x)}{p(d)}\right] \tag{1}$$
>
> where $X$ is the input sample, $D$ is the input representation, i.e., model's output. The *leave one out* upper bound (UB) [2] can be utilized to conservatively bound MI:
>
> $$
> I(X ; D) \leq \mathbb{E}\left[\frac{1}{N} \sum_{i=1}^{N} \log \frac{p\left(d_{i} \mid x_{i}\right)}{\frac{1}{N-1} \sum_{j \neq i} p\left(d_{i} \mid x_{j}\right)}\right]=I_{\mathrm{UB}} \tag{2}
> $$
>
> Following the information theory fundamentals, by applying the conventional Probably Approximately Correct (PAC) of GE bound, we can obtain the bound on GE as:
>
> $$
> \mathrm{GE}<\sqrt{\frac{\log (|\mathcal{H}|)+\log (1 / \delta)}{2 N_{\mathrm{trn}}}} \tag{3}
> $$
>
> where $|\mathcal H|$ is the hypothesis-class cardinality and $N_\mathrm{trn}$ is the number of training examples. For the synthetic data, $N_\mathrm{trn}=|\mathcal C_\mathrm{syn}|$, while for the full data, $N_\mathrm{trn}=N_\mathrm{full}$. The confidence parameter, denoted by $\delta$ and ranging between 0 and 1, specifies the likelihood that the bound remains consistent with respect to the chosen $N_{\mathrm{trn}}$ training samples.
>
> According to the property of deep neural networks [1], the cardinality of the hypothesis space reduces to $|\mathcal{H}| \approx 2^{|\mathcal T|}$ where $|\mathcal{H}|$ is the number of class-homogeneous clusters that the backbone network distinguishes.  An estimate for the number of clusters can then be obtained by $|\mathcal T| \approx 2^{H(X)} / 2^{H(X \mid Z)}=2^{I(X ; Z)}$.
>
> The ability of Input Compression Bound (ICB) [3, 4] is to predict changes in GE under different dataset interventions, it then can be formulated as:
>
> $$
> \mathrm{GE}_{\mathrm{ICB}} < \sqrt{\frac{2^{I(X ; D)} + \log (1 / \delta)}{2 N\_{\operatorname{trn}}}} \tag{4}
> $$
>
> Thus, we can have the generalization error bound for the condensed data as:
>
> $$
> \mathrm{GE}^{\mathrm{syn}}\_{\mathrm{ICB}}<\sqrt{\frac{2^{I(X ; D)}+\log (1 / \delta_\text{syn})}{2 | \mathcal C\_{\operatorname{syn}}|}} \tag{5}
> $$
>
> where the generalization error bound for full dataset is $\mathrm{GE}^{\mathrm{full}}\_{\mathrm{ICB}}<\sqrt{\frac{2^{I(X ; D)}+\log (1 / \delta_\text{full})}{2 N\_{\operatorname{full}}}}$.
>
> We will include a more detailed and well-organized theoretical analysis into the revised version of our paper.
>
> ### References
>
> [1] Angus Galloway, Anna Golubeva, Mahmoud Salem, Mihai Nica, Yani Ioannou, and Graham W. Taylor. "Bounding generalization error with input compression: An empirical study with infinite-width networks." Transactions on Machine Learning Research (TMLR) 2022.
>
> [2] Ben Poole, Sherjil Ozair, Aaron Van Den Oord, Alex Alemi, and George Tucker. On Variational Bounds of Mutual Information. In Proceedings of the 36th International Conference on Machine Learning, volume 97 of Proceedings of Machine Learning Research, pp. 5171–5180. 2019.
>
> [3] Naftali Tishby. Information Theory of Deep Learning, 2017.
>
> [4] Ravid Shwartz-Ziv, Amichai Painsky, and Naftali Tishby. Representation Compression and Generalization in Deep
> Neural Networks. OpenReview, 2019.

---

> > ### Comment · Reviewer_4huB · 2023-08-17
> > **Response to the Rebuttal**
> >
> > I would like to express my appreciation to the authors for their detailed and insightful response to my review. Their effort in providing additional theoretical support has indeed addressed my initial concerns and has significantly enriched my understanding of the paper's contributions.
> >
> > The authors' response has provided a clear and compelling explanation of how their approach fits into the broader landscape of related methods. The clarification regarding the theoretical underpinnings of their method, along with the comparisons to existing techniques, has solidified my confidence in the novelty and importance of their work.
> >
> > I commend the authors for their diligence and thoughtful responses, and I look forward to the continued refinement and impact of their work in the field.

---

> > > ### Author Response · Authors · 2023-08-18
> > > **Thanks for your comments**
> > >
> > > We sincerely thank you for your acknowledgment and uplifting comments. Wishing you a wonderful day.

---

### Author Rebuttal · Authors · 2023-08-09

Dear Reviewers,

We would like to express our gratitude for your insightful feedback and comments, which have been helpful in updating and enhancing our submission. We kindly invite you to review our author rebuttal so that we may address any further questions you may have or clarify any points that remain unclear. In summary, our rebuttal mainly includes the following:

* We carry out theoretical analysis of generalization error bound through Input Compression Bound (ICB) and Mutual Information (MI) between the original/condensed input and the final layer representations. (Reviewer 4huB)
* We present results on ConvNet for dataset distillation, a sensitivity analysis with respect to $\alpha_{BN}$, and additional experiments on increasing resolution and increasing the number of classes. (Reviewer QGHe)
* We break down the whole training time consumption at each stage to demonstrate the superior efficiency of our proposed approach. (Reviewer RJ8s)
* We provide a detailed explanation of BN-ViT architecture. (Reviewer RJ8s)
* We provide the comparison between our dataset condensation and directly sampling the same scale of images. (Reviewer RJ8s)
* We expand our analysis to include additional comparisons with the baseline approach. (Reviewer MBn1)
* We present the comparison and discussion of the cross-architecture generalization ability. (Reviewer 7q3V)

We hope our responses can adequately address your concerns. We will integrate all the comments presented in the rebuttal into the revised paper, and we sincerely appreciate your valuable feedback.

Best,
Authors

---

### Decision · Program_Chairs · 2023-09-21

**Decision:**

Accept (spotlight)

**Comment:**

The paper was received well by the reviewers. Particularly, the reviewers commended
- clear presentation,
- novel framework,
- impressive performances,
- and extensive experiments.

Some of the issues, such as
- limited theoretical analysis,
- ablation studies,
- and cross-architecture generalisation
are effectively addressed in the rebuttal.

Our final decision is an acceptance. We kindly ask the authors to include the additional contributions made throughout the rebuttal process in the final version.